# Patterns and controls of inter-annual variability in the terrestrial carbon budget

Barbara Marcolla[1], Christian Rödenbeck[2], Alessandro Cescatti[3]

[1]Fondazione Edmund Mach, IASMA Research and Innovation Centre, Sustainable Agro-ecosystems and Bioresources Department,38010 San Michele all'Adige, Trento, Italy
[2]Max Planck Institute for Biogeochemistry, 07745 Jena, Germany.
[3]European Commission, Joint Research Centre, Directorate for Sustainable Resources, I-21027 Ispra (VA), Italy

*Correspondence to*: Alessandro Cescatti (alessandro.cescatti@ec.europa.eu)

**Abstract.** The terrestrial carbon fluxes show the largest variability among the components of the global carbon cycle and drive most of the temporal variations in the growth rate of atmospheric $CO_2$. Understanding the environmental controls and trends of the terrestrial carbon budget is therefore essential to predict the future trajectories of the $CO_2$ airborne fraction and atmospheric concentrations. In the present work, patterns and controls of the inter-annual variability (IAV) of carbon Net Ecosystem Exchange (NEE) have been analysed using three different data-streams: ecosystem level observations from the FLUXNET database (La Thuille and 2015 releases), the MPI-MTE bottom-up product resulting from the global up-scaling of site-level fluxes, and the Jena CarboScope Inversion, a top-down estimate of surface fluxes obtained from observed $CO_2$ concentrations and an atmospheric transport model. Consistencies and discrepancies in the temporal and spatial patterns and in the climatic and physiological controls of IAV were investigated between the three data sources. Results show that the global average of IAV at FLUXNET sites (~120 gC $m^{-2}$ $y^{-1}$), quantified as the standard deviation of annual NEE, is observed to peak in arid ecosystems and to be almost six times larger than the values calculated from the two global products (15 and 20 gC $m^{-2}$ $y^{-1}$ for MPI-MTE and Jena Inversion, respectively). The two data-driven global products show that most of the temporal variability observed in the last three decades is due to yearly anomalies, whereas the temporal trends explain only about 15% of the variability in the MPI-MTE product and 20% in the Jena Inversion product. Both at site level and at global scale, the IAV of NEE is driven by the gross primary productivity and in particular by the cumulative carbon flux during the months when land acts as a sink. Altogether these results offer a broad view on the magnitude, spatial patterns and environmental drivers of IAV from a variety of data sources that can be instrumental to improve our understanding of the terrestrial carbon budget and to validate the predictions of land surface models.

## 1 Introduction

Atmospheric $CO_2$ concentration has been constantly increasing since the Industrial Revolution, and has caused a corresponding rise of 0.85 °C in the global air temperature from 1880 to 2012 (IPCC, 2013). Since the 1960s, terrestrial ecosystems have acted as a considerable sink of atmospheric $CO_2$, reabsorbing about one quarter of anthropogenic emissions (Friedlingstein et al., 2010; Le Quéré et al., 2014). The growth rate of atmospheric $CO_2$ concentration is characterized by a large inter-annual variability (IAV), which mostly results from the variability of the $CO_2$ net ecosystem exchange (NEE) on land (Bousquet et al., 2000; Le Quere et al., 2009; Yuan et al., 2009). Multisite synthesis confirms that a large inter-annual variability in NEE is a common feature at all flux sites around the world (Baldocchi, 2008; Baldocchi et al., 2001). The reason why the IAV is so large is that NEE results from the small imbalance between two larger fluxes: the photosynthetic uptake of $CO_2$ (Gross Primary Production, GPP) and the respiratory release of $CO_2$ (Total Ecosystem Respiration, TER). As a consequence, even minor variation in either of the two fluxes can cause large variations in their difference.

It has been long debated which of GPP or TER controls the spatial and temporal variability of NEE. Several studies have ascribed inter-annual variability in NEE to variability in either GPP (Ahlstrom et al., 2015; Janssens et al., 2001; Jung et al.,

2011, 2017; Stoy et al., 2009; Urbanski et al., 2007) or TER (Morgenstern et al., 2004; Valentini et al., 2000) or both (Ma et al., 2007; Wohlfahrt et al., 2008b). GPP and TER show comparable ranges of IAV, typically larger in absolute terms than that observed for NEE due to the temporal correlation between the two gross fluxes (Richardson et al., 2007). Given that photosynthesis and respiration may respond differently to environmental drivers (Luyssaert et al., 2007; Polley et al., 2008), the interpretation of climate impacts on the variability of NEE requires the understanding of the relation between the IAV of NEE and that of GPP and TER (Polley et al., 2010).

The environmental factors driving the IAV of NEE ($IAV_{NEE}$) include: climate, physiology, phenology, natural and anthropogenic disturbances (Marcolla et al., 2011; Richardson et al., 2007; Shao et al., 2015). Understanding the spatio-temporal variability of NEE and its controlling mechanisms is essential to assess the vulnerability of the terrestrial carbon budget, to evaluate the land mitigation potentials and to quantify the ecosystem capacity to store carbon under future climatic conditions (Heimann and Reichstein, 2008). Besides, quantifying inter-annual variability in NEE is a prerequisite for detecting longer-term trends or step changes in flux magnitude in response to climatic or anthropogenic influences and identifying its drivers (Cox et al., 2000; Lombardozzi et al., 2014).

The temporal dynamic of NEE has been addressed in numerous studies, based on either ''top-down'' approaches, which primarily focuses on aircraft atmospheric budgets (Leuning et al., 2004), tower based boundary layer observations (Bakwin et al., 2004) and tracer transport inversion (Baker et al., 2006; Gurney et al., 2002; Rödenbeck et al., 2003), or on ''bottom-up'' methods that rely on data-driven gridded products derived from the up-scaling of flux data (Jung et al., 2011, 2017; Papale et al., 2015; Papale and Valentini, 2003) or process-based biogeochemical models that simulate regional carbon budgets (Desai et al., 2008, 2007; Mahadevan et al., 2008).

Despite the broad literature on the subject, very few examples of IAV analysis based on multiple data streams are available in the literature (Desai et al., 2010; Pacala, 2001; Poulter et al., 2014). In the present study patterns and controls of the inter-annual variability of NEE have been analysed using three different data streams: ecosystem level data from the FLUXNET database, the MPI-MTE bottom-up product resulting from the statistical up-scaling of in-situ flux data (le Maire et al., 2010) and the Jena CarboScope Inversion top-down product, which estimates land (and ocean) fluxes from atmospheric $CO_2$ concentration measurements and atmospheric transport modelling (Rödenbeck et al., 2003). In particular, this analysis aims to: i) assess the magnitude and the spatial pattern of IAV of NEE ($IAV_{NEE}$), ii) investigate the role of key climatic variables, like temperature and precipitation, in driving the spatial pattern of IAV and iii) identify the role of photosynthesis and respiration as sources of $IAV_{NEE}$. Finally, the consistencies and discrepancies among the different data products are analysed and critically evaluated.

## 2 Materials and Methods

### 2.1 Datasets

Data at ecosystem scale were retrieved from two releases of the FLUXNET dataset, namely LaThuile (http://fluxnet.fluxdata.org/data/la-thuile-dataset/) and 2015 (http://fluxnet.fluxdata.org/data/fluxnet2015-dataset/). These datasets contain half-hourly data of carbon dioxide, water vapour and energy fluxes that are harmonized, standardized and gap-filled. Time series of NEE and of the component fluxes GPP and TER, together with air temperature and precipitation, were used in the present analysis. Flux data have the advantage to represent direct observations of in-situ IAV, however at most sites the time series are still too short for a proper analysis of the temporal variability of NEE (Shao et al., 2015). For this reason only sites with a minimum of five years of observations and an open data distribution policy were selected. A subset of 89 sites satisfied the two criteria, among which 27 evergreen needle-leaf forests, 5 Evergreen Broadleaf Forests, 12 deciduous broad-leaf forests, 6 mixed forests, 12 grasslands, 8 croplands, 6 sites counting closed and open shrublands, 7 wetlands and 6 sites counting savannahs and woody savannahs.

At global scale, two sources of gridded data were used: a "bottom-up" data product, namely the MPI-MTE product (Jung et al., 2009) and, as "top-down" product, the Jena CarboScope $CO_2$ Inversion (Rödenbeck et al., 2003). The MPI-MTE dataset is built with a machine learning technique (model tree ensemble, MTE) to upscale in space and time the flux observations from the global network of eddy covariance sites (FLUXNET) integrated with climate and remote sensing data for the time period 1982-2011 (Jung et al., 2009). Global maps for GPP and TER at 0.5° spatial resolution and monthly temporal resolution were used, while NEE fields were calculated as difference between the gross fluxes. This product has become a reference dataset to evaluate process-oriented land models and remote sensing estimates of primary productivity, despite the uneven distribution of eddy covariance sites on which it is trained. It integrates a large amount of in-situ measurements, remote sensing and meteorological observations using a machine learning technique and has been proved to reproduce well spatial patterns and seasonal variability of fluxes (Jung et al., 2009). On the other hand the product has some shortcomings: for instance the effects of land management, land use change and $CO_2$ fertilization are not represented. The MPI-MTE has been recognized to underestimate the inter-annual variability of carbon fluxes. These limits may be due to the missing representation of some key determinants of IAV like changes in soil and biomass pools, disturbances (e.g. fires), ecosystem age, management activities and land use history. Finally, the lag between external forcing and ecosystem response is not represented in the product (Jung et al., 2011).

To derive surface fluxes, the Jena CarboScope Inversion combines modelled atmospheric transport with high-precision measurements of atmospheric $CO_2$ concentrations. Atmospheric transport is simulated by a global three-dimensional transport model driven by meteorological data. For consistency with the MPI-MTE product, monthly averaged NEE land fluxes from the s81_v3.6 version of the product were used here, at a spatial resolution of 5° x 3.75°. The Jena Inversion is particularly suited for the analysis of temporal trends and variability since it is based on a temporally constant observation network (14 atmospheric stations for the version s81_v3.6). Weaknesses of the Jena Inversion product are linked i) to the sparse density and biased spatial distribution of the sampling network, whose geometry affects the flux estimates in a systematic way, ii) to data gaps, iii) to measurement errors since the calculation is based on $CO_2$ data only, while atmospheric carbon comes also from CO and VOC, and iv) to potential systematic errors of the transport model that cannot be assessed.

As the inversion estimates the total land flux, being calculated as the difference between the total surface flux and prescribed anthropogenic emissions, it includes also $CO_2$ emissions from fires in addition to NEE. For improving the consistency with the other two datasets (MPI-MTE and FLUXNET) that do not account for fire emissions, we subtracted fire emissions from the inversion estimates using an harmonized combination of the products RETRO (Schultz et al., 2008) for the period 1982-1996 and GFED4 (Van Der Werf et al., 2010) for the period 1997-2013. RETRO is a global gridded data sets (at 0.5° spatial resolution) for anthropogenic and vegetation fire emissions of several trace gases, covering the period from 1960 to 2000 with monthly time resolution. GFED4 combines satellite information on fire activity and vegetation productivity to estimate gridded monthly fire emissions at a spatial resolution of 0.25 degrees since 1997. RETRO and GFED4 were harmonized using the overlapping years (1997-2000) to calculate calibration coefficients as the ratio of GFED4 to RETRO for latitudinal bands of 30°. The RETRO time series was then multiplied by these coefficients and the resulting time series of fire emissions was finally subtracted from the land flux of the Jena Inversion. It is worth noting that the remaining flux from the inversion is the sum of land use change emissions and NEE while the MPI-MTE does not account for the land use change flux.

In order to analyse the role of climatic drivers on the inter-annual variability, global maps of temperature and precipitation were used. Gridded air temperatures were obtained from the Climatic Research Unit (CRU) at the University of East Anglia at monthly time scale and 0.5°x0.5° spatial resolution, based on an archive of monthly mean temperatures provided by more than 4000 weather stations (Jones et al., 2012). Precipitation fields were obtained from the GPCC product at 0.5° and monthly time step (Schneider et al., 2014) . This product is based on a large dataset of monthly precipitation from

more than 85,000 stations and is provided by NOAA/ESRL PSD (Boulder, Colorado, USA). The MODIS MCD12C1 land cover product (Friedl and Brodley, 1997) was used to classify the land pixels and to calculate statistics by plant functional type. MCD12C1 provides the dominant land cover types at a spatial resolution of 0.05° using a supervised classification algorithm that is calibrated using a database of land cover training sites. Product resolutions were harmonized using the aggregate function of the raster R-package.

## 2.2 IAV Analysis

The inter-annual variability of NEE was estimated as the standard deviation of annual NEE values generated by trend and residuals, computed on time windows of 12 months shifted with a monthly time step (Luyssaert et al., 2007; Shao et al., 2015; Yuan et al., 2009) and calculated with the same methodology for the three data-streams used in the analysis. Average values of IAV for plant functional type (PFT) were determined using the PFT classification of FLUXNET sites and the MCD12C1 product (aggregated at the appropriate spatial resolution using the dominant PFT) for the MPI-MTE and Jena Inversion. Map grid-cells were also classified according to mean annual temperature and precipitation, and the mean value of $IAV_{NEE}$ and normalized $IAV_{NEE}$ were calculated for each climate bin.

For the two gridded products, which provide a 30 year long time series (1982-2011), the IAV was partitioned in two components, namely the variance explained by the temporal trend and that due to annual anomalies (Ahlstrom et al., 2015). For this purpose a linear model was fitted on the time series at each pixel, and the determination coefficient of the regression was used to measure the fraction of variance explained by the trend, whereas its complement to one was the fraction of variance due to anomalies.

The spatial correlation between IAV and climatic drivers (air temperature and precipitation) was analysed at global scale for the MPI-MTE by calculating the spatial correlation coefficient between the temporal standard deviation (IAV amplitude) of NEE and the average annual temperature or precipitation in moving spatial windows of 15°x11.5° (which means 31x21 pixels for MPI-MTE). The latitudinal averages of these correlation coefficients were calculated for latitudinal bands of 30°. This analysis was not replicated on the Jena Inversion because at fine scale the spatial variability of the fluxes in this product is mainly controlled by the prior estimates. In fact, the optimization algorithm of the inversion spatially allocates the fluxes proportionally to the prior; hence grid cells with higher productivity will change more if compared to cells with a lower prior.

Finally, in order to identify which process between photosynthesis and respiration drives $IAV_{NEE}$, for FLUXNET and MPI-MTE linear regressions between NEE and GPP and NEE and TER were fitted for each site/pixel and the difference between the determination coefficients of the two linear regressions was computed. Since GPP and TER cannot be derived from inversion products, we performed a similar analysis using NEE of the Carbon Uptake Period (CUP, sum of negative monthly NEEs) and of the Carbon Release Period (CRP, sum of positive monthly NEE), as proxies of GPP and TER for all the three data streams. Also in this case NEE was linearly regressed with $NEE_{CUP}$, and $NEE_{CRP}$ to detect which of the two processes drives the variability of NEE. $IAV_{NEE}$ and IAV controls were also analysed in a climatic space defined by mean annual temperature and precipitation. Finally, annual anomalies of the two global products used in the present analysis were compared with the estimates derived from the Global Carbon Project (GCP) (Quéré et al., 2016).

## 3 Results and Discussion

### 3.1 IAV patterns

The spatial pattern of inter-annual variability for the three data sets is resumed in Figure 1. The IAV of NEE at individual Fluxnet sites ranges from 15 to 400 gC m$^{-2}$y$^{-1}$ and shows an average of 130 gC m$^{-2}$y$^{-1}$. On average the most northern sites show a lower temporal variability both in Europe and in North America (Fig. 1a). A global map of $IAV_{NEE}$ is shown also for

MPI-MTE (Fig. 1b) and Jena Inversion (Fig. 1c) at the original spatial resolutions of the two products. The observed range of IAV is similar for the two gridded products and substantially lower than that observed at site level, probably due to the spatial averaging of the land fluxes that dampens the temporal variability. The mean global value of IAV is in fact 15 and 20 gC m$^{-2}$ y$^{-1}$ for MPI-MTE and Jena Inversion, respectively, and hence about one sixth of the site level IAV. The two gridded products confirm the decreasing trend of IAV toward northern latitudes observed at flux sites. A general decrease of IAV$_{NEE}$ at higher latitude for both ENF and DBF was also observed by Yuan et al. (2009) although for none of the two PFTs these trends were significant.

In terms of IAV, the two global products show a reasonable qualitative correspondence for North America and Eurasia, whereas they disagree for South America, with MPI-MTE showing a minimum of IAV in the humid Tropics, where the inversion product shows a large variability. MPI-MTE in particular shows maximum values along the Eastern coast of South America while the Jena Inversion shows an almost opposite pattern. A similar behaviour is observed also in Africa, where the top-down product shows a maximum in central Africa while MPI-MTE shows a minimum in the Congo basin and higher values in arid zones like Sahel and South Africa. These discrepancies could, on the one hand, be ascribed to the limits of the bottom-up approach in dealing with the low seasonality of the fraction of absorbed radiation (FaPAR) in evergreen broadleaf forests, given the relevance of this predictor in the MPI-MTE estimates. A second reason for the discrepancy could be due to the CO$_2$ emissions from land use change that is particular relevant in some tropical areas but are not accounted in the MPI-MTE estimates. On the other hand, the fine-scale estimates of the inversion are largely determined by the a-priori weighting pattern, which has been chosen proportional to time-mean NPP (from the LPJ model) as a vegetation proxy (Rödenbeck et al., 2003). As the atmospheric data can only constrain larger-scale patterns comparable to the distances between the stations, this means that IAV will be locally higher where mean NPP is high, and vice versa.

As far as the Northern Hemisphere is concerned, a good correspondence is observed in western Eurasia, while some discrepancies are observed in other zones; for example MPI-MTE shows a large IAV in India, probably driven by the changes in FaPAR related to agricultural intensification, which is not emerging from the inversion product that has little observational constraint in this area. To summarize, the spatial pattern of IAV in the two products better agrees in the Northern Hemisphere for temperate and cold temperate zones, whereas for the southern Hemisphere, and in particular for the humid evergreen forests, they show a poor match. In general it has to be considered that both the MPI-MTE product and the Jena Inversion are driven by data from surface networks that are very limited in the Tropics and Southern Hemisphere and, therefore, these observation-driven estimates are under-constrained in those areas. These results highlight that for achieving more robust and consistent estimates of the terrestrial carbon fluxes it is of key importance to increase the availability of atmospheric and ecosystem flux observations in the Tropical region, either establishing new sites where the network is sparse or improving the sharing of data where the monitoring stations are available but not connected to global networks (e.g. flux stations in Amazonia).

The results presented in the maps of Figure 1 are summarized in the climate space in Figure 2. The left panels show that peak values of IAV are located in different climate regions for the two gridded products (temperate humid for MPI-MTE, and tropical humid for Jena Inversion). These results highlights that top-down and bottom-up estimates do not agree on the main sources of temporal variability in the terrestrial carbon budget and call for more investigation to pin down the reasons for these large discrepancies. Given that the standard deviation of NEE increases with the primary productivity at the Fluxnet sites (Figure 3), in Figure 2 (right column) we normalized IAV of both MPI-MTE and Jena Inversion by the average GPP of the specific climate bin from the MPI-MTE. Normalization using GPP (which is always positive) offers a more robust metric of relative IAV if compared to normalization with NEE (that spans across zero). Figure 2 reports in each climate bin either the mean IAV (left column) or the ratio of the mean IAV and GPP (right column), since this metric is less sensitive to outliers than the mean of ratios and gives more weight to points with larger fluxes. The normalized IAV shows a consistent pattern between the three different data products, with a clear decreasing trend at increasing temperature and precipitation

(i.e. increasing productivity). Ultimately arid regions seems to have the higher relative variation in land carbon fluxes, in accordance with previous findings (Ahlstrom et al., 2015). Interestingly the two gridded products show slightly different climatic location for the peak in relative IAV, with MPI-MTE pointing to warm arid regions whereas Jena Inversion points to cold arid systems.

The dependency of $IAV_{NEE}$ on GPP and on $NEE_{CUP}$ is reported in Figure 3 for the three datasets. Both for Fluxnet and the Jena Inversion, IAV is increasing with both GPP and $NEE_{CUP}$. On the contrary, the $IAV_{NEE}$ in the MPI-MTE dataset peaks at intermediate values of GPP and $NEE_{CUP}$, even if this trend is not evident in the Fluxnet data from which the MPI-MTE product is derived. As stressed previously, MPI-MTE seems to underestimate the temporal variability of evergreen tropical forests both in South America and Africa, where the highest values of GPP and $NEE_{CUP}$ are observed and where on the contrary the inversion shows high values of IAV. We think that this mismatch is due to the prominent role that FaPAR has in the MTE approach. In fact, canopy greenness is particularly stable in the tropical humid forests, generating this unusual pattern of low relative IAV in regions of high productivity. These contrasting results for key regions like the Amazon and the Congo basin confirm the large uncertainty of the IAV estimates in areas with limited observational constraints. In these regions, climate sensitivities derived from estimates of the inter-annual variability of the terrestrial carbon budget have therefore to be carefully interpreted (Fang et al., 2017).

The importance of the spatial scale of analysis on the $IAV_{NEE}$ has been explored for both Fluxnet sites and the global products (i.e. MPI-MTE and Jena Inversion) (Figure 4). The two global products show a good agreement at the native Inversion resolution (5°x3.75°) and at global level, when only one global value is retrieved by spatially averaging all the pixels of the original maps. For the MPI-MTE product, the observed IAV is decreasing regularly at decreasing map resolution. On the contrary, the Jena Inversion shows a rapid descent followed by a stabilisation. Fluxnet sites and their aggregation at increasing distance also show a decreasing IAV with higher values compared to the global products. The slope of the lines in Figure 4 reveals the degree of spatial compensation between anomalies (steeper slopes are generated by stronger compensation and therefore lower spatial coherence), which leads to a decrease of $IAV_{NEE}$ at the increase of the spatial extent of the observations. Among the three products MPI-MTE shows the gentler slope and therefore the larger spatial coherence of the anomalies. This is possibly due to the missing representation of land disturbances in the MTE methodology, which may ultimately lead to an overestimation of the spatial coherence in the land $CO_2$ flux anomalies.

The fractions of $IAV_{NEE}$ generated either by temporal trends or by annual residuals are summarized in Fig. 5 for the two global gridded products. For MPI-MTE, more than 80% of the IAV is explained by residuals at all latitudes. Only in limited zones like Congo and Western Amazonia, MPI-MTE shows a relative minimum in the importance of residuals, but this global product might underestimate the total variability in these zones (see Fig. 1b). Residuals explain the largest share (between 62 and 90%, average 77%) of the temporal variability also in the Jena Inversion, with a higher relevance of trends in the southern hemisphere. The inversion product shows several hotspots of trend-driven variability, like South Africa, South America and northern Eurasia that is indeed reported as an area of increasing productivity in the last decades. In the interpretation of these results it is important to consider that MPI-MTE is generated by the statistical upscaling of Fluxnet data, using climate and FaPAR as predictors. This methodology relies on the assumption of a constant ecosystem response to climate drivers and for this reason the product cannot reproduce the influence of some environmental factors (e.g. increasing $CO_2$ concentration or nitrogen deposition) that may alter these responses and that are not reflected in input variables like climate or FaPAR. On the contrary, inversion products do not make any assumption on the climate dependence of ecosystem functioning, but include also emission from land management and land use change that may hide or emphasize the NEE trends. In summary, it is important to notice that, despite the important climate trends, in the last 30 years the temporal variability of the land carbon balance has been driven by annual residuals, confirming the dominant role of climate variability on the terrestrial C budget (Le Quéré et al., 2014).

For the two gridded products the analysis of IAV (either in terms of absolute $IAV_{NEE}$ or normalized with $NEE_{CUP}$) was disaggregated by plant functional type (Figure 6). The analysis in terms of absolute $IAV_{NEE}$ shows that savannahs and woody savannahs (WSAV-SAV) are the PFTs characterized by the larger IAV and variability within the PFT. This was found both for the MPI-MTE and the Jena Inversion product and confirms the results of a recent study (Ahlstrom et al., 2015) in which semi-arid ecosystems were found to account for the largest fraction (39%) of the global IAV in net biome productivity. This variability was found to be significantly related to the length of the growing season (Ma et al., 2007) and is driven by the uncertainty in water supply in arid systems. In terms of normalized IAV the two gridded products show different behaviours, CSH-OSH being the most variable PFT for MPI-MTE while the inversion data report a higher variability for EBF and WSAV-SAV. As observed at pixel scale in Figure 1, even at PFT level the results obtained from Fluxnet sites show a higher variability than the gridded products. In general at Fluxnet sites IAV is proportional to ecosystem productivity (Fig 4) with the maximum values observed in EBF, DBF-MF and CRO-GRA and the minimum in WET. The large value of IAV observed in GRA-CRO is presumably also affected by the potential large impact of management in these ecosystems that can either reduce (e.g. by irrigation) or increase the climate-induced variability (e.g. by changing crops or fertilization schemes, etc.). In general the disaggregation of $IAV_{NEE}$ by PFTs shows rather similar results between the two gridded products, both in terms of magnitude and distribution. The largest difference is observed in the evergreen broadleaf forests whose absolute and relative variability is much larger in the inversion, possibly as a result of the intensive disturbances that have occurred in these ecosystems over the last decades and that are not captured with the MTE methodology.

### 3.2 Climate dependence of IAV

The climatic dependence of the spatial variability of $IAV_{NEE}$ at global scale for the MPI-MTE product (Figure 7) shows a clear pattern with positive correlations in temperature-limited areas at northern latitudes, and negative temperature dependence in water-limited zones (Braswell et al., 1997). These observations agree with Reichstein et al. (2007), which report that GPP shifts from soil water content to air temperature dependency at around $52°$ N. These opposite temperature dependences will probably lead to future contrasting changes in IAV. In fact, under a global warming scenario, the northern latitudes will be characterized by a larger sink (Zhao and Running, 2010) but also by a larger temporal variability, while arid zones like the Mediterranean basin, the Middle East Australia and the Sub-Saharan Africa will probably experience a reduction in IAV linked to large-scale droughts and consequent reduction in primary productivity (Ciais et al., 2005). Concerning precipitation the MPI-MTE product show more complex spatial patterns with negative correlation in the humid tropics, temperate Europe and South-East USA and positive correlation elsewhere.

The climate dependencies of IAV are further separated between the variability due to trends and anomalies (Fig. 7 right column). The two components of $IAV_{NEE}$ mostly show an agreement in the sign of the climatic controls, meaning that the environmental drivers have the same effects on trends and anomalies and therefore on the long and short time scales. This is a relevant finding because it supports the use of IAV to investigate medium term climatic responses. In general anomalies show a higher correlation than trends, probably due to the larger magnitude of the variance attributed to this component. In conclusion, the spatial patterns shown in the maps of Fig. 7 and the agreement between the two components of IAV reported in the barplots indicate that the temperature controls of IAV of NEE is in general the same as for the primary productivity (i.e. positive in colder biomes and negative in warmer regions), while the contrasting results observed for precipitation suggest that the role played by water availability on the spatial and temporal variability is unclear, probably because of the temporal correlation between precipitation and temperature anomalies, as shows by Jung et al. (2017). The analysis of the climate drivers of IAV was not performed for the Jena Inversion because for this product local variation in IAV are heavily driven by the prior estimates of NPP and therefore results have limited sensitivity to the atmospheric constraints.

### 3.3 Physiological drivers of IAV

290      An improvement in the mechanistic understanding of $IAV_{NEE}$ can be achieved by partitioning the net flux in its two components: GPP and TER. Partitioned fluxes are available for Fluxnet sites and for derived products like MPI-MTE, while they cannot be derived from atmospheric inversions. For this latter product the fluxes during the Carbon Uptake Period (CUP; NEE<0) and during the Carbon Release Period (CRP; NEE>0) were used in this analysis as proxies of GPP and TER, respectively.

295      To investigate how good these proxies are, the ratios TER/GPP during CUP and GPP/TER during CRP were analysed at Fluxnet sites and for each pixel of the MPI-MTE product and averaged by PFT (Figure 8a). As far as the MPI-MTE product is concerned, TER ranges from 55 to 78% of GPP during the CUP while GPP is 56 to 80% of TER during the CRP, hence on average about two-thirds of the signal come from GPP (TER) in the CUP (CRP). These ratios show a certain variability among PFTs, with ENF having the larger imbalance between the two fluxes and the lowest ratio TER/GPP during CUP (due 300 to the strong seasonality of GPP in this PFT), while the two fluxes are not so well partitioned for EBFs (ratio ~0.8) that are characterized by a long growing season with consistently large fluxes of GPP and TER. The other PFTs show an average ratio value of ~0.65 both in CUP and CRP. In summary, it can be inferred that NEE during CUP is dominated by the signal of GPP, while NEE during CRP is dominated by TER even though to a smaller extent, as it emerges from the frequency distributions in Figure 8bc calculated from the MPI-MTE product. The distribution of the ratio TER/GPP during the CUP is 305 in fact narrower and peaks at a value of 0.7, while a broader distribution is observed for the GPP/TER ratio during the CRP. As expected there is a larger spread in the composition of NEE during CRP across the World, and this is linked to the larger variability in the seasonality of GPP that may actually go to zero in the dormancy season, while TER is always positive.

     In order to identify which of the gross fluxes controls the variability of the net land flux we assessed the fraction of variance ($R^2$) of NEE explained by GPP or TER (for MPI-MTE and Fluxnet) and CUP or CRP (for all products). Results 310 reported in figure 9 show the difference of the determination coefficients between the two regressions (NEE versus GPP or TER; NEE versus $NEE_{CUP}$ or $NEE_{CRP}$) and are used to determine which component dominates the inter-annual variation of NEE. Blue zones in figure 9 are regions where $IAV_{NEE}$ is driven by photosynthesis (GPP or CUP), being the difference $R_{GPP}^2 - R_{TER}^2$ (or $R_{CUP}^2 - R_{CRP}^2$) positive, while in the red zones $IAV_{NEE}$ is mainly controlled by respiration (TER or CRP). Figure 9a shows that, in most of the land area, the $IAV_{NEE}$ is driven by GPP both at Fluxnet sites and for the MPI-MTE 315 product. The same data products show an even clearer dominance of $NEE_{CUP}$ on IAV (Fig. 9b). The Jena Inversion product shows that, although most of the globe is $NEE_{CUP}$ driven, there are quite a few areas that are weakly CRP driven like eastern US, arid regions in Africa and the Amazon basin, probably because these areas are estimated to be $CO_2$ sources in this inversion and therefore NEE is dominated by $NEE_{CRP}$ (data not shown). When latitudinal profiles are considered, all the products show that GPP and $NEE_{CUP}$ control the temporal variability of yearly NEE more than TER or $NEE_{CRP}$ (le Maire et 320 al., 2010). Results shown in the global maps of Figure 9 are represented in the climatic space in Figure 10. Map pixels were classified according to mean annual temperature and precipitation. For each climate bin the difference between the determination coefficients for NEE vs GPP and TER is reported. Across the whole climate space, IAV retrieved from the MPI-MTE product is mostly controlled by CUP and GPP, although the difference in $R^2$ in the case of GPP and TER is low. The Jena Inversion on the contrary shows climate areas where IAV is CRP driven, especially in intermediate-high 325 temperature classes. Similar results have been reported across several PFT by Yuan et al., (2009) and Ahlstrom et al. (2015) using Fluxnet site data and MTE products. A higher correlation of IAV with GPP rather than with TER in deciduous forests has been reported also by Barr et al. (2002) and Wu et al. (2012). These results suggest that ecosystem fluxes during the CUP, and in particular photosynthesis more than respiration, are consistently controlling the inter-annual variability of NEE at all the spatial scales for both "bottom-up" and "top-down" data products (Janssens et al., 2001; Luyssaert et al., 2007; le 330 Maire et al., 2010; Urbanski et al., 2007; Wohlfahrt et al., 2008a; Wu et al., 2012; Yuan et al., 2009). These results highlight that temporal variations of photosynthesis and of ecosystem $CO_2$ exchange during the carbon uptake period are therefore

driving the short-term climate sensitivity of the global carbon cycle consistently across different regions and climates. The possibility to interpret these short-term responses as long-term potential impacts of climate change is therefore to be disputed; given the limited role that respiration appears to play in modulating the rapid reactions of the terrestrial biosphere to environmental drivers.

In order to place our analysis in a broader context, global annual values of the gridded products used in the present analysis have been finally compared with the estimates of the Global Carbon Project (Figure 11). At annual timescale Jena Inversion shows an excellent agreement with the GCP, and this is not surprising since GCP land fluxes are estimated as residual term from the atmospheric $CO_2$ budget and are therefore not completely independent from the Jena Inversion product. On the other hand, this analysis highlights how the MTE bottom-up approach is barely correlated with the top-down estimates, both in term of trend and of anomalies. These discrepancies may be partially explained by the missing representation of land disturbances (land use change, land management) in the MTE product.

In conclusion, this study assessed the temporal variability of the terrestrial C budget with three different datasets to diagnose common patterns and emerging features. Some discrepancies between data-products have emerged, in particular in the Tropics were a chronic deficiency of atmospheric and ecosystem observations is severely limiting the accuracy of large-scale assessments. On the other hand, several important global features have been identified and confirmed by the different products like: i) the dominant role played by photosynthesis in the short-term variability of the land carbon budget, ii) the high relative IAV in water limited eco systems and iii) the dependence of IAV on spatial scales and ecosystem productivity. Ultimately, the variability of the land fluxes observed in the recent decades proved to be extremely valuable to investigate the controlling mechanisms, the sensitivity and vulnerability of the terrestrial C balance to climate drivers.

**AKNOWLEDGEMENTS**

This study was supported by the JRC project AgForCC nr.442. The MCD12C1 product was retrieved from the online Data Pool, courtesy of the NASA EOSDIS Land Processes Distributed Active Archive Center (LP DAAC), USGS/Earth Resources Observation and Science (EROS) Center, Sioux Falls, South Dakota. This work used eddy covariance data acquired and shared by the FLUXNET community, including these networks: AmeriFlux, AfriFlux, AsiaFlux, CarboAfrica, CarboEuropeIP, CarboItaly, CarboMont, ChinaFlux, Fluxnet-Canada, GreenGrass, ICOS, KoFlux, LBA, NECC, OzFlux-

TERN, TCOS-Siberia, and USCCC. The FLUXNET eddy covariance data processing and harmonization was carried out by the ICOS Ecosystem Thematic Center, AmeriFlux Management Project and Fluxdata project of FLUXNET, with the support of CDIAC, and the OzFlux, ChinaFlux and AsiaFlux offices.

GPCC Precipitation data provided by the NOAA/OAR/ESRL PSD, Boulder, Colorado, USA, from their Web site at http://www.esrl.noaa.gov/psd/

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

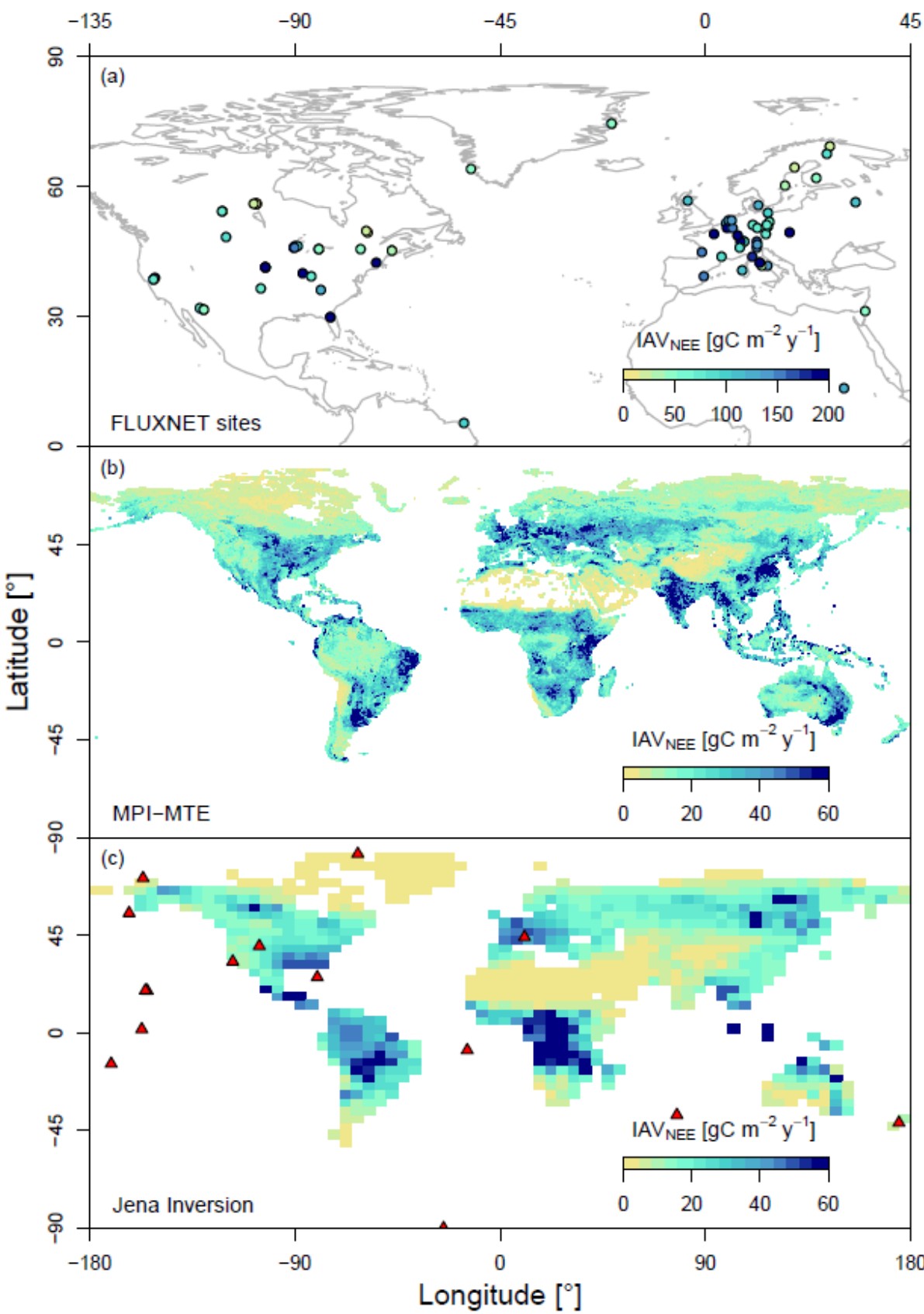

**Figure 1: Spatial distribution of NEE standard deviation used as a measure of inter-annual variability (IAV$_{NEE}$). Results are reported for a) Fluxnet sites with at least 5 years of observations, b) for the MPI-MTE NEE product and c) Jena Inversion product s81_v3.6, red triangles represent the CO$_2$ concentration measurement sites.**

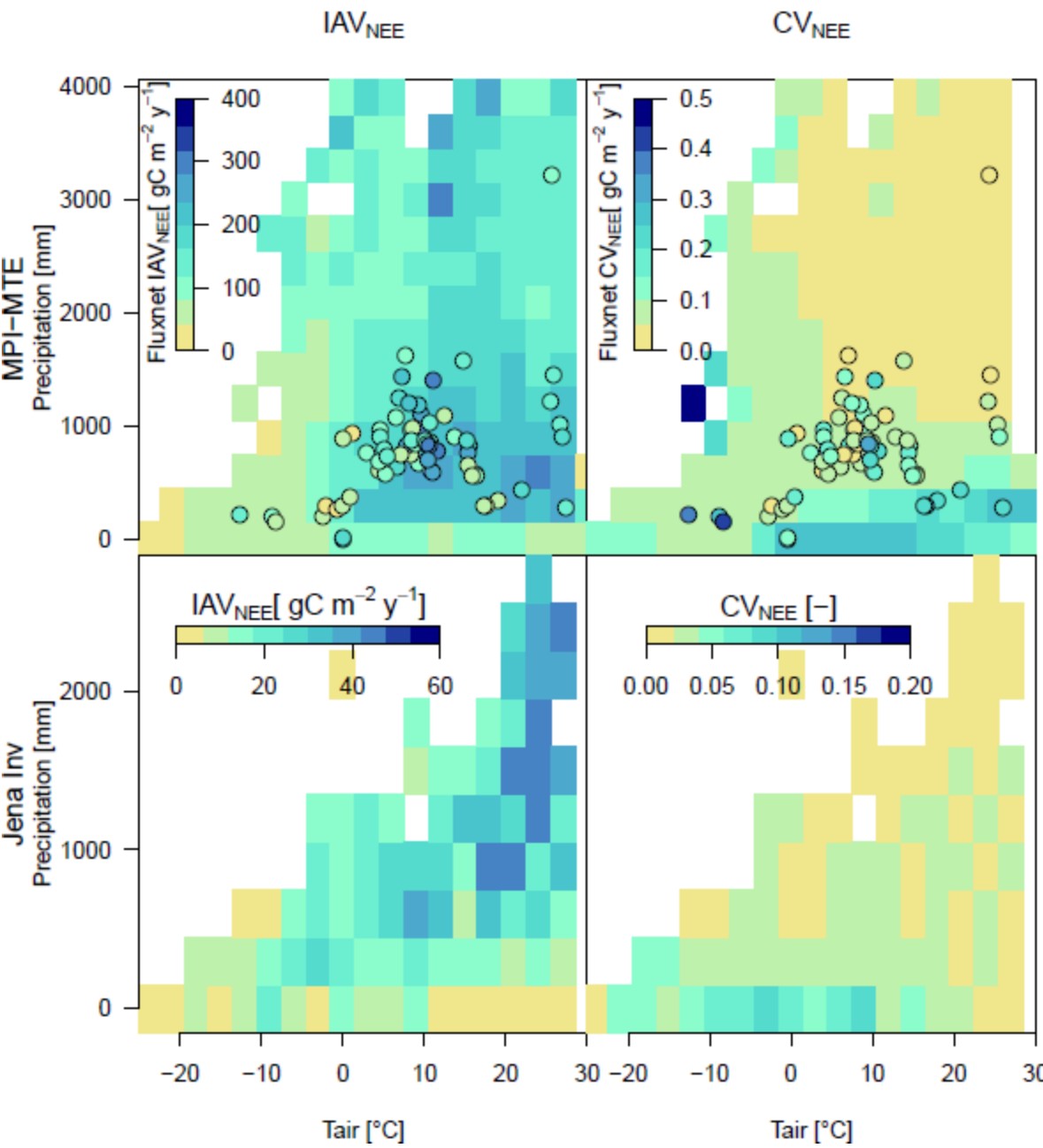

**Figure 2: IAV$_{NEE}$ (left panels) and normalized IAV$_{NEE}$ (CV$_{NEE}$, right panels) plotted in a Temperature-Precipitation space, for MPI-MTE (top panels) and Jena Inversion (bottom panels). Dots represent Fluxnet site values.**


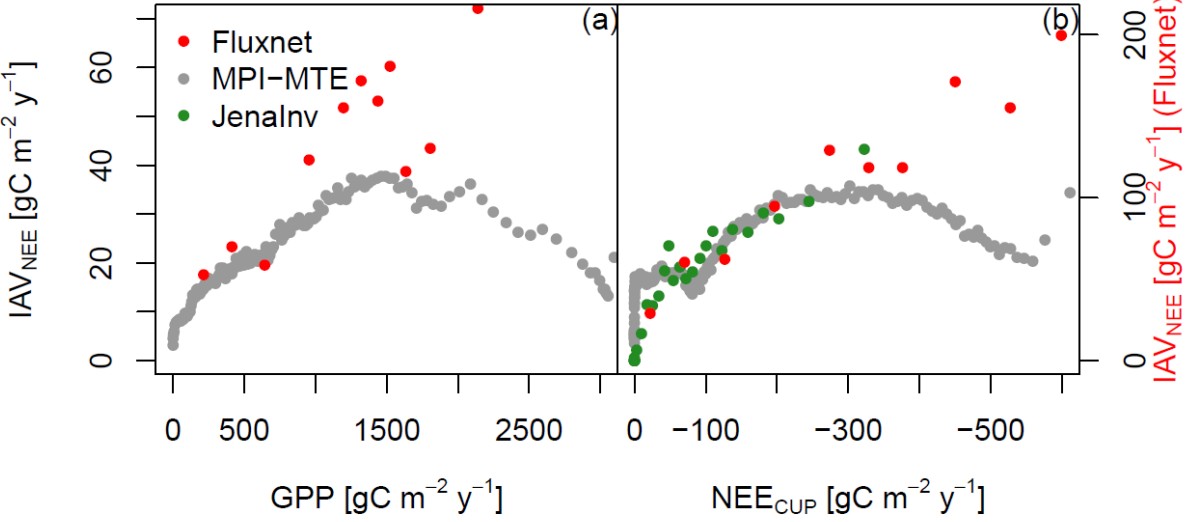

**Figure 3: Dependency of standard deviation of NEE on GPP and NEE$_{CUP}$ . Results are reported for Fluxnet sites (red dots, different y scale on the right), for the MPI-MTE NEE (black dots) and Jena Inversion product (green dots)**

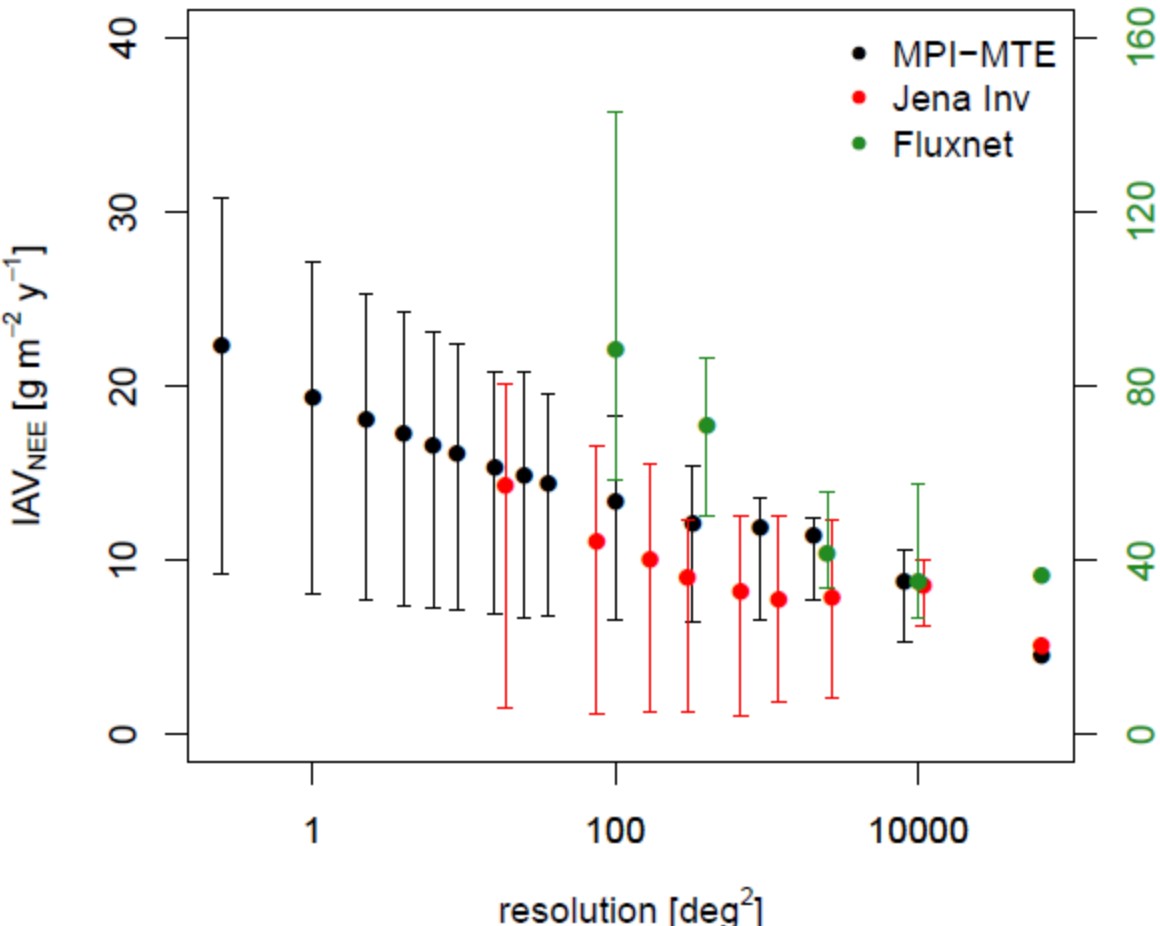

**Figure 4: Dependence of IAV$_{NEE}$ on map resolution for Fluxnet sites (green dots), MPI-MTE (black dots) and Jena Inversion (red dots). Error bars represent the 25% and 75% quantiles of the IAV in the aggregated sites/pixels.**

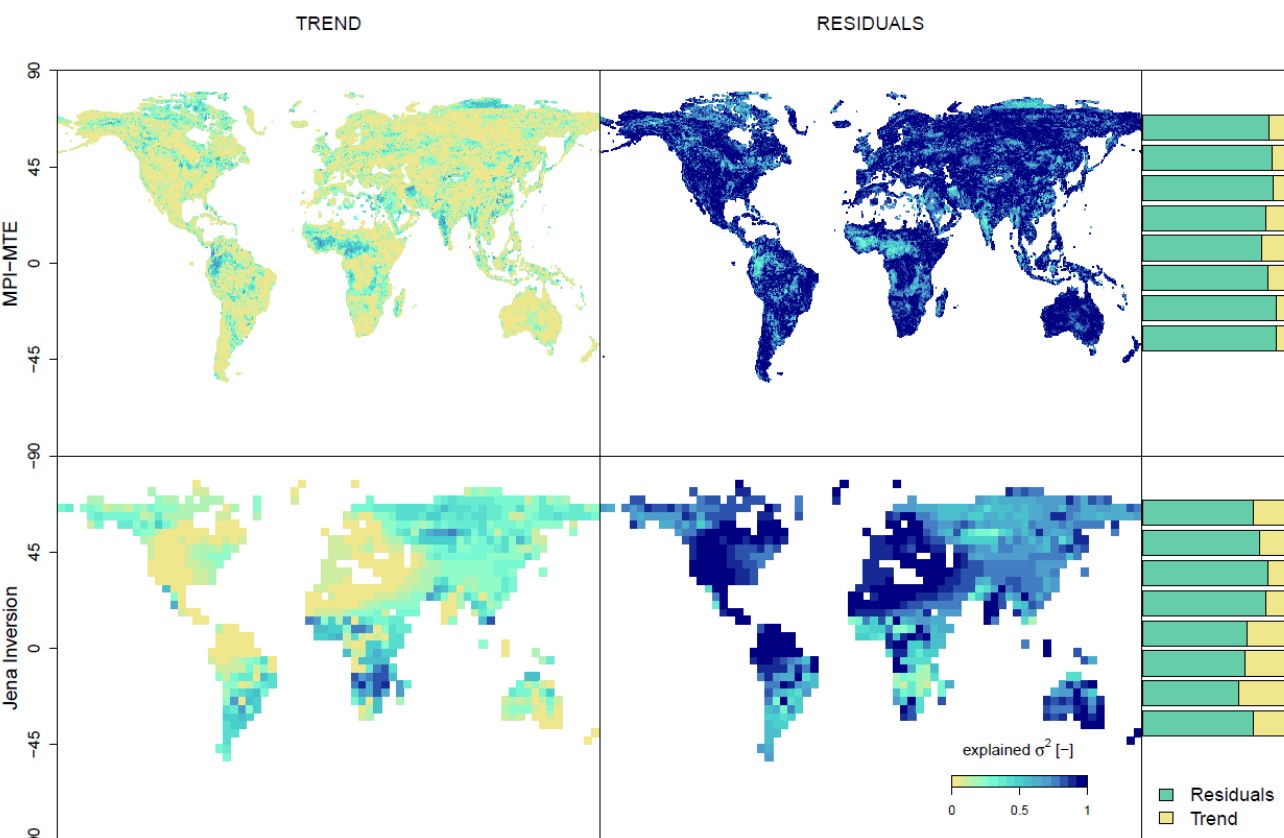

**Figure 5: Maps of the fraction of NEE variance explained by temporal trends and anomalies for MPI-MTE NEE and Jena Inversion; latitudinal band (15°) averages of the fractions are reported in the bar plots.**

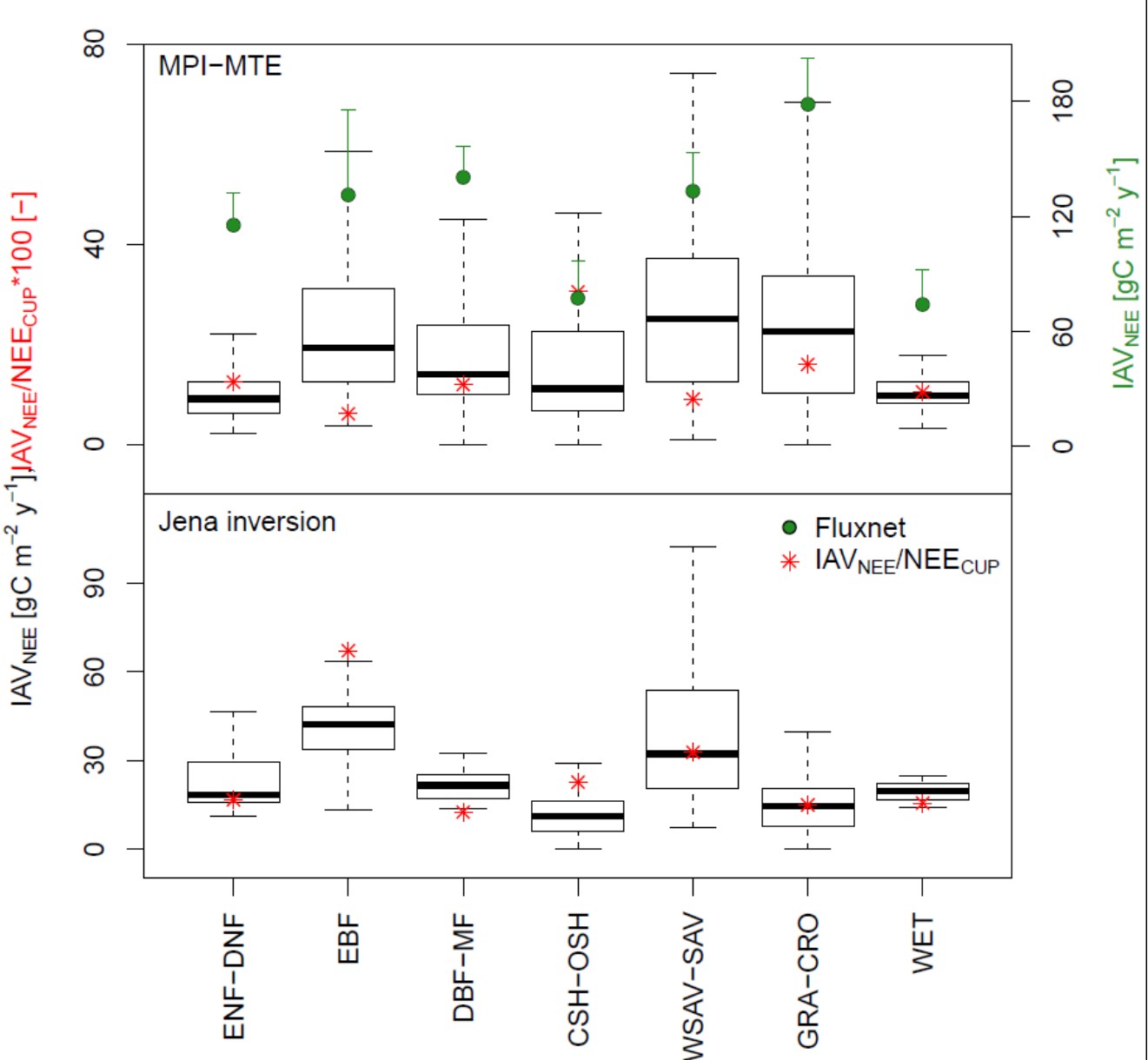

Figure 6: Boxplot of NEE standard deviation averaged in PFT classes for MPI-MTE NEE and Jena Inversion, green dots represent observations at Fluxnet sites (different y scale on the right) plotted with their standard error. PFT classes are grouped as follows: evergreen needleleaf forests and deciduous needleleaf forests (ENF-DNF), evergreen broadleaf forests (EBF), deciduous broadleaf forests and mixed forests (DBF-MF), closed and open shrublands ( CSH-OSH), woody savannahs and savannahs (WSAV-SAV), grasslands and croplands (GRA-CRO) and wetlands (WET).

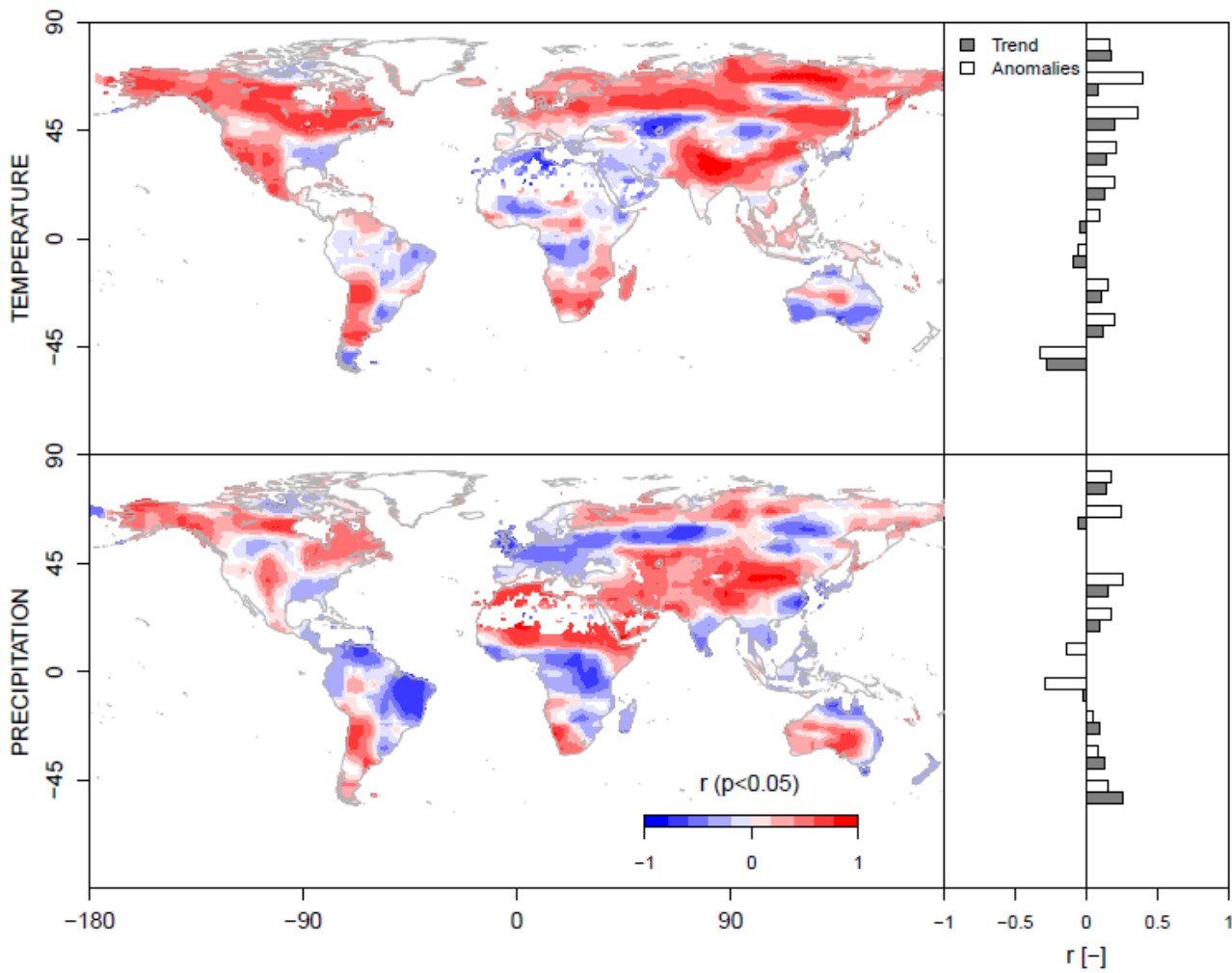

**Figure 7: Climatic drivers of the spatial variability of NEE standard deviation. The left panels show maps of the spatial correlation coefficients (within moving spatial windows of 15°x11.5°) of interannual NEE amplitude versus time-mean temperature and precipitation for the bottom up product MPI-MTE. Pixels with non-significant correlation are left white. The barplots on the right show latitudinal averages of the correlation coefficients of NEE trend and anomalies versus temperature and precipitation.**

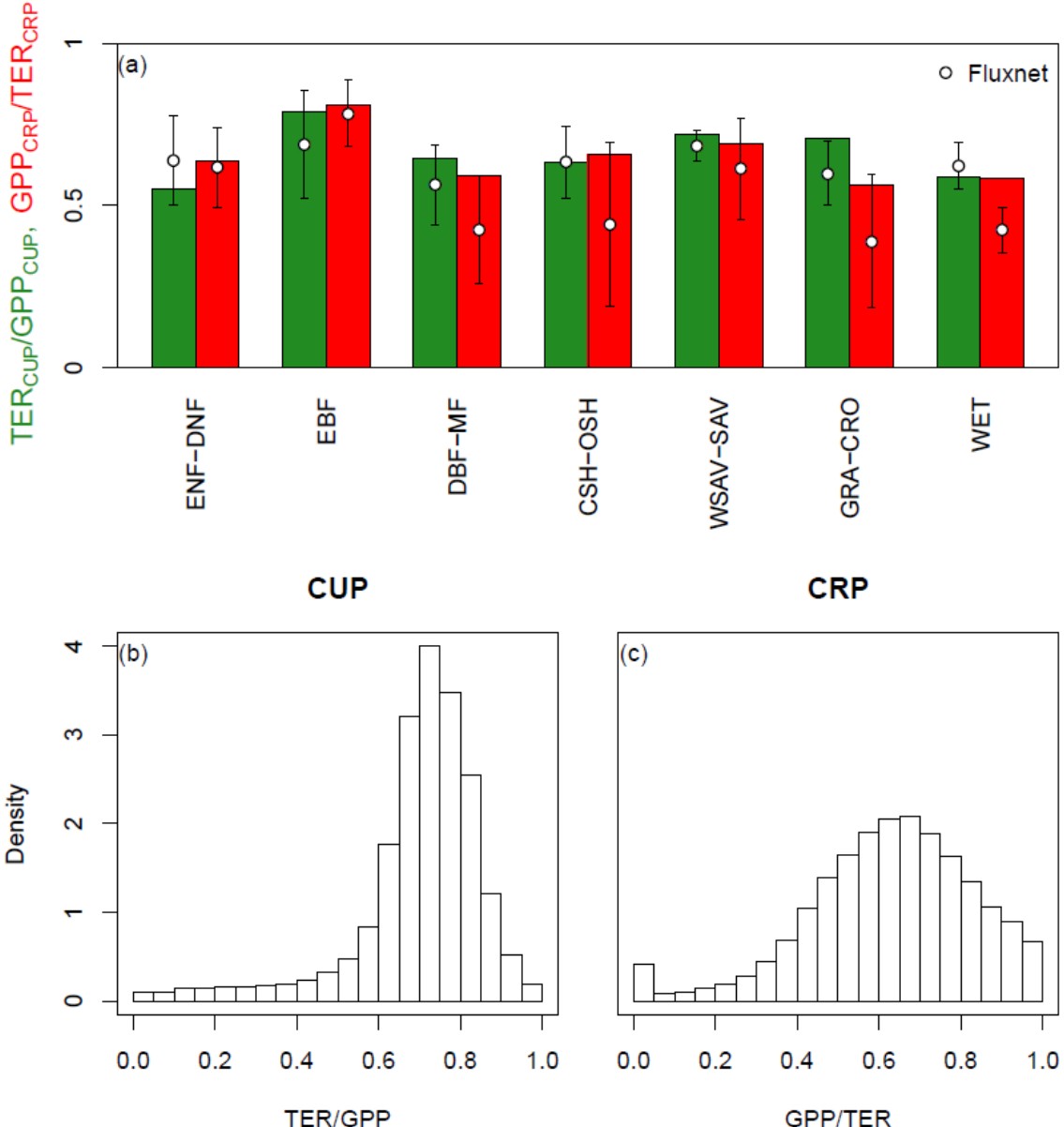

**Figure 8: Bar plot of the ratio TER/GPP for the Carbon Uptake Period (CUP, green bars) and of the ratio GPP/TER during the Carbon Release Period (CRP, red bars), these values were calculated for the MPI-MTE product, dots refer to Fluxnet sites. Averages of yearly values are represented together with their standard deviation. The global frequency distributions of the ratios obtained from the MPI-MTE product are reported in the histograms at the bottom.**


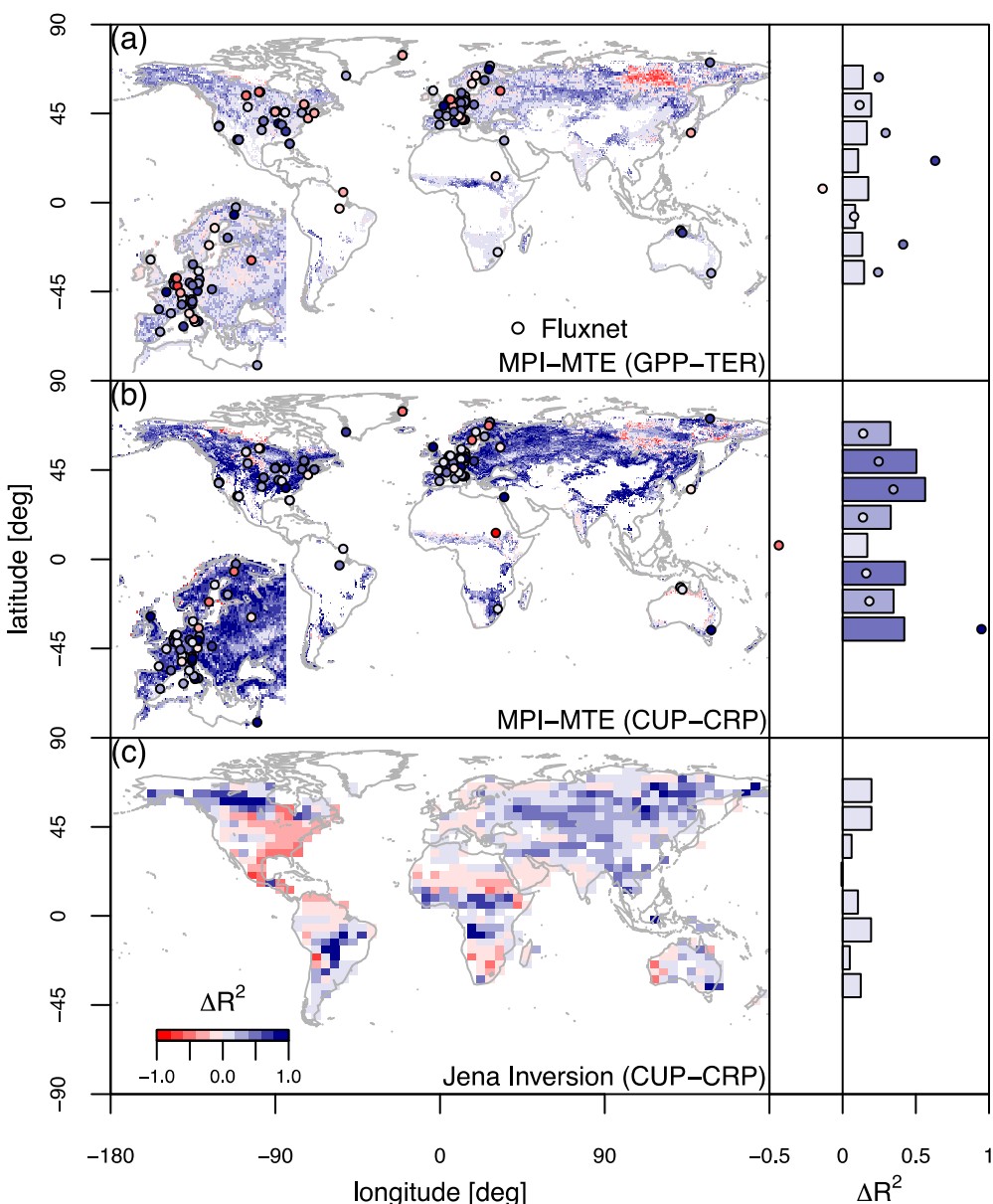

**Figure 9: Control on IAV by GPP-TER and Carbon Uptake Period (CUP) - Carbon Release Period (CRP), expressed as the difference of the determination coefficients for Fluxnet sites with at least 5 years of observations (dots), MPI-MTE NEE and Jena Inversion. Latitudinal averages are reported for latitudinal classes of 15 degrees. Inset maps show an enlarged plot of Europe.**



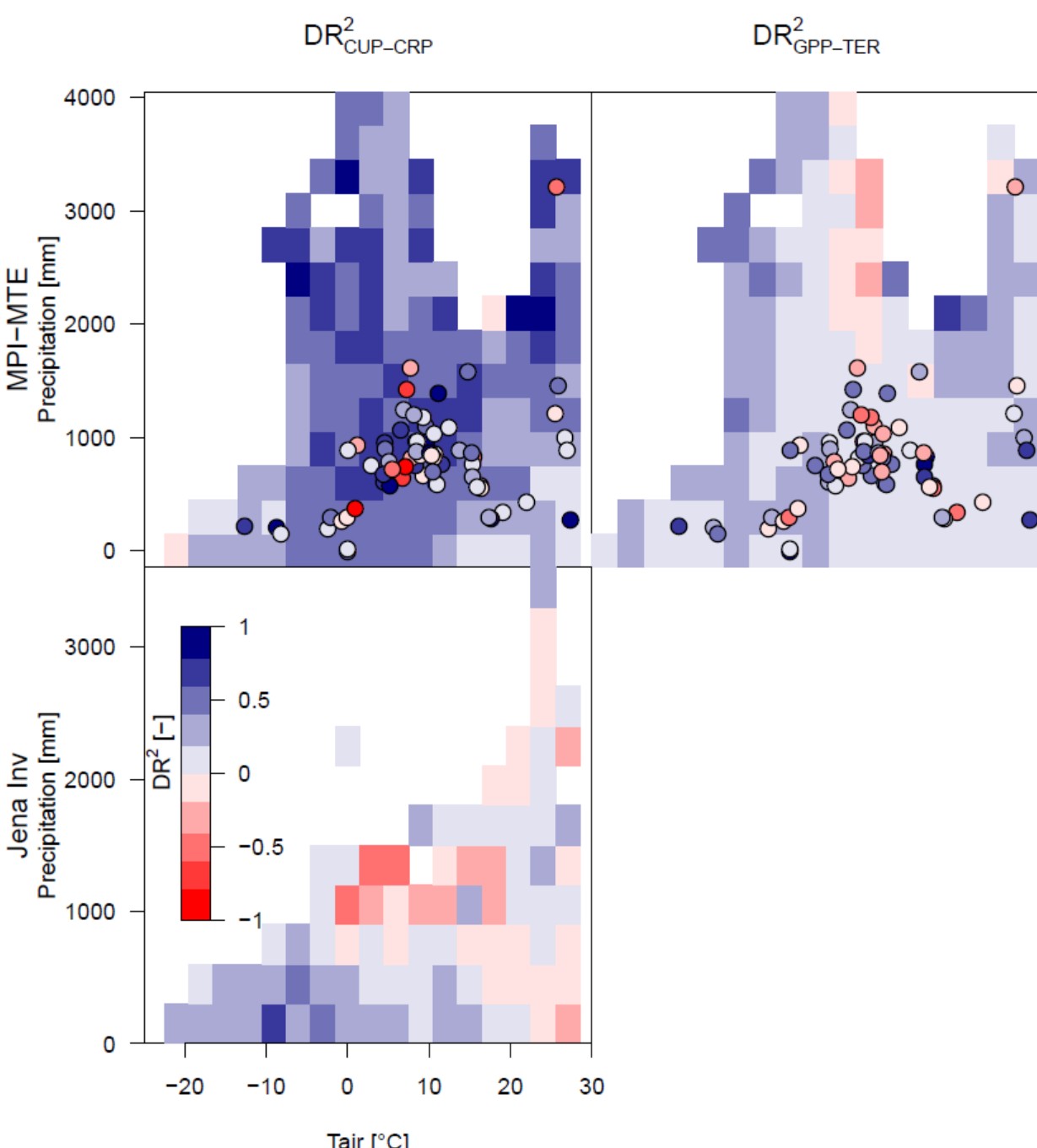

**Figure 10: Control on IAV by GPP-TER and Carbon Uptake Period (CUP) - Carbon Release Period (CRP), expressed as the difference of the determination coefficients plotted in a Temperature Precipitation space. The two top panels refers to MPI-MTE while the bottom panel to Jena Inversion, dots refer to Fluxnet sites.**


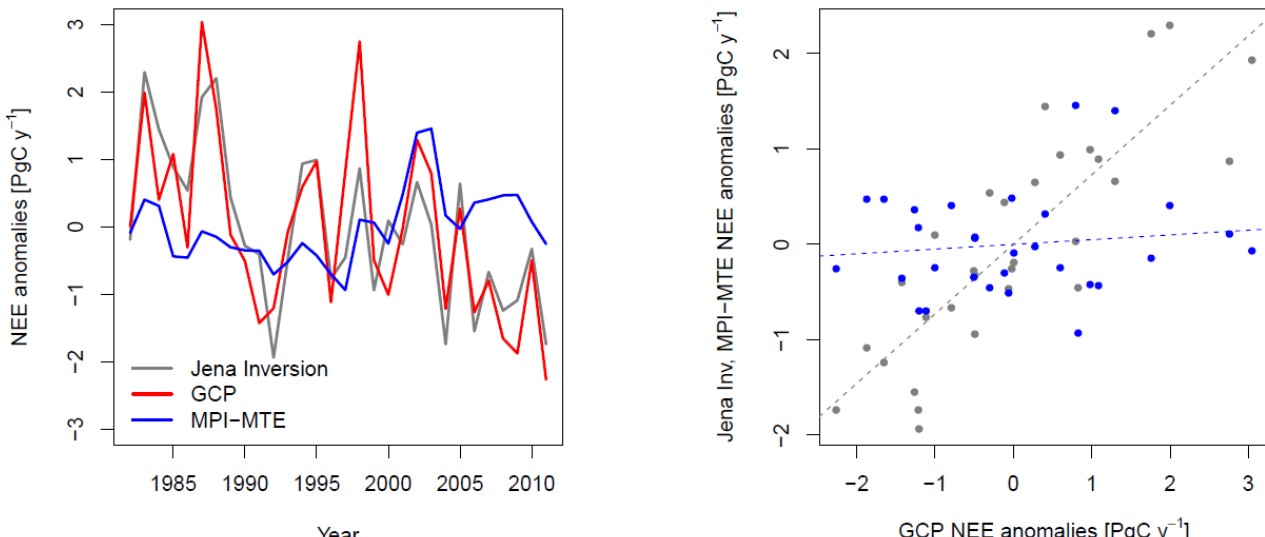

Figure 11: Comparison of the annual NEE anomalies between Jena Inversion, MPI-MTE and the Global Carbon Project data. Time series of annual anomalies are reported in the left panel while regressions of MPI-MTE and Jena Inversion values are reported in the right panel versus annual anomalies of the Global Carbon Project Data