# Peer review of "Patterns and controls of inter-annual variability in the terrestrial carbon budget"

_Biogeosciences, 2017_

## Referee Comment (RC1) · N. Krakauer (Referee) · 26 Jan 2017

This work compares interannual variability (IAV) in land NEE (and its GPP and respiration components) for 1982-2011 between eddy covariance flux tower data, a global empirical upscaling of this data (MPI-MTE), and an inversion based on a small number of station $CO_2$ time series.

**1  Major comments**

1. All three datasets are of dubious accuracy in representing interannual variability. The annual totals computed from eddy covariance sum much larger fluxes of opposing signs with likely systematic biases, especially in nighttime. The empirical

upscaling was found to have relatively weak performance in representing inter-annual variability in a synthetic data experiment (without even accounting for any measurement or representativeness error in the training set) reported by Jung et al. (2009), for which the absence of soil moisture as a predictor is given by them as one reason. The inversion estimate, as the authors point out, is dominated at sub-continental scales by the (reasonable) prior assumption that variability scales with modeled NPP, and it probably contains little actual information from the $CO_2$ time series at those scales. Could it makes sense to run the inversion with a more 'flat' prior, or a prior based on the MPI-MTE IAV, to get different IAV estimates?

2. Figure 4 shows the dependence of median(?) IAV on resolution for the two grid-ded products. I wonder if something like this could be done with the available Fluxnet stations as well, for example with the help of a variogram (mean covari-ance of deseasonalized NEE time series as a function of inter-station distance). This could help in deciding whether the lower IAV in the gridded products com-pared to Fluxnet is only because of the difference in spatial scale or is more intrinsic.

3. I didn't see any analysis of to what extent the IAV between the three products is actually in phase (i.e. the correlation of the deseasonalized NEE time series between the datasets). It would probably be relevant to show this.

4. Also, forest inventories and crop yield statistics provide more reliable direct mea-surements of (at least above-ground) NPP and its IAV in many countries, po-tentially with rather good spatial coverage. Would there be any way to compare these to the IAV in the data sets reported here?

In summary, this is a valuable exercise, but I would like to see fuller discussion of the uncertainties, limitations, and potential checks and improvements noted above.

**2  Minor points**

1. The element "carbon" is not capitalized (title and line 287).

2. Line 25: no comma before "that"

3. Figure 1c: It would be good to show the station network on the map.

4. "Anomalies" sounds strange as a description of the IAV residuals from linear trend shown in Figure 5 and discussed in the text. Perhaps there is a better term.

5. The Jung et al. (2009) citation should be to the final paper, not the discussion paper.

6. Formatting in the bibliography needs to be fixed, e.g. for Morgenstern et al. (2004) and others.

---

## Referee Comment (RC2) · Anonymous Referee #2 · 26 Feb 2017

General comments

This manuscript describes an analysis of the interannual variability (IAV) of net ecosystem exchange (NEE), based on three different and complementary, but not completely independent, data sources: FLUXNET, MPI-MTE (a bottom up gridded product derived from FLUXNET), and a top-down CO2 inversion-based product. This is an interesting topic, given the importance of understanding the controls on year-to-year terrestrial-atmosphere carbon exchange, and appropriate for Biogeosciences. The ms is reasonably well written, and the technical analysis generally strong. If there's nothing terribly surprising in the results, it's a useful analysis of both likely patterns of, and controls on, the IAV of NEE, and the strengths and weaknesses of the different NEE products themselves.

There are some weaknesses. Some areas of the text, and a critical point or two in the methods, are unclear. Neither the MPI-MTE nor the inversion products seem ideal for this kind of IAV analysis, although I recognize that this is all there is to work with; still, the authors should address this. In addition, the conclusion should be re-done or removed; on a related note, the strengths and weaknesses of these NEE data products might be better, and more succinctly, summarized based on the analyses performed.

These are, however, relatively minor points, as I found this a strong ms overall.

Specific comments

————————————

1. Lines 118-120: not as clear as it should be. Interannual variability computed with a 12-month window? How is this possible, as that's only 1 year?

2. L. 171-172: move to figure caption, or methods

3. L. 197: "area of"

4. L. 241-243: unclear

5. L. 250-: separating paragraphs, or indenting their first lines, would make this easier to read

6. L. 286-: these aren't conclusions, just a recapitulation of results; remove

7. Figure 2: Rain (in axis title) or Precipitation (in caption)?

---

## Referee Comment (RC3) · Anonymous Referee #3 · 9 Mar 2017

Marcolla and co-authors look at three different NEE products, compare the inter-annual variability (IAV) of each, and relate these measurements of IAV back to climate variability and contributions of plant productivity and ecosystem respiration. The study is nice, but incomplete in some ways or confusing in others. These suggestions are intended to strengthen the impact of the paper.

————Broad comments————-

I feel like the paper is missing the bigger take home message I was looking for, to the globally (or Fluxnet) integrated anomalies in NEE match up with 1) each other and 2) anomalies in the land C sink the global carbon project (Le Quéré et al. 2014; these data are available in a downloadable spreadsheet at http://www.globalcarbonproject.org/carbonbudget)

Since the paper is ostensibly about inter-annual variability in the terrestrial C cycle (NEE) what aren't all data products detrended first (these are weak responses anyway caused by different assumptions made with each approach)? Then the authors would be better able to address the IAV (or anomalies) which seem to be the focus of the paper.

What climate or weather data are used in MTE or the Jena inversion. Presumably neither used CRU (temperature) and GPCC (precipitation), as the authors of this paper chose to do? Thus, are analyses of climate drivers on IAV of NEE actually really just comparisons of distinct climate reanalysis products? Also, why not use the CRU precipitation product for consistency with the temperature data being used?

Much of the text in section 3 is heavy on the results with little discussion and interpretation of the key findings. Although some sections do communicate broader statements about the findings (e.g. lines 197-206), similar thoughtful development of ideas should be included throughout this section

Why aren't correlations of IAV with site – level or global-scale climate drivers shown for Fluxnet or Jena inversion products?

I'm unclear what value is communicated by the calculation of CUP and CRP and would suggest removing these analyses from the paper. The finding that temperate and boreal systems have a stronger seasonal cycle in their $CO_2$ drawdown seems obvious from atmospheric $CO_2$ growth curves. Instead, if the purpose of these analysis is to "identify the role of photosynthesis and respiration as sources of IAV_NEE" (line 67), then it seems much more straightforward to just look at the IAV (or anomalies) of GPP and TER from the Fluxnet and MTE products directly. Then they could be correlated with climate drivers too? For example, at high latitudes do GPP and TER show strong temperature sensitivities, with anomalies GPP outpacing TER in warm years? Conversely, are Tropical GPP anomalies largely temperature related too, whereas TER shows less inter annual variability & climate sensitivity?

—————-Minor comments & Technical corrections—————-

I'm not used to seeing citations in the abstract. Is that the format for this journal?

I'm used to seeing ecosystem respiration referred to as ER, but maybe the authors are used to using different conventions?

Line 55 This single paragraph is a single sentence consisting of a very long list of NEE estimation approaches. Why not break this into a sentence about each approach and discuss strengths/ weaknesses of each?

Line 67 organization of objectives i), ii), and iii) don't align with the organization of methods and sections 3.1, 3.2 and 3.3. Can the objective reflect the broader layout of the paper?

I'd suggest Line 73 are "LaThuile and 2015" two distinct references?

It's not clear if or how data were re-gridded (e.g. [1] subtracting finer scale RETRO and GFED4 fire fluxes from the Jena inversion, or [2] for temperature and precipitation in Fig. 2).

More broadly, is subtracting for fire fluxes even necessary? Do the 14 observations extrapolated to this global product even 'see' the effect of forest fires? Don't the atmospheric inversion products the global carbon project implicitly see the effects of these fires? If so, why should the be subtracted out here?

Line 80. There are enough abbreviations in the text already. Are these needed too? Their use in lines 210-219 makes the text very hard to follow.

Line 109 Air should not be capitalized.

'Jena inversion' or 'Jena Inversion' should be used consistently throughout the text.

Were any lagged correlations explored to see if climate variability affected NEE in the subsequent season / year?

Standard deviation and IAV are used interchangeably throughout the manuscript, but I think they mean the same thing? If so, just one term should be used for consistency. If they are different, it should be clarified in the text.

Line 133 I have no idea what this means "the difference between the two determination coefficients was computed" or where this analysis is presented (Fig 9)? More broadly, I'm unclear how / why the authors tried to infer something about GPP and TER from the inversion product.

Line 171. Why was IAV normalized using GPP estimates and not NEE, the later giving a real coefficient of variation (CV; grid cell standard deviation NEE / mean grid cell NEE). This should be clarified both in the text and caption. Also, shouldn't grid cell CV be calculated first, and then averaged over each climate bin?

Line 180 & Fig. 3 I am unclear what insight this figure provides to the manuscript and it's sparingly discussed in the text. It's used to justify the CV calculation in Fig. 3 (line 173), but as this is a standard statistical approach I'm not sure it's warranted? As such, should the display item just be removed?

Line 200. It seems like 'trends' in IAV should be driven mainly by environmental presses like atmospheric $CO_2$ concentrations or broad-scale / chronic N deposition inputs. By contrast, climate variability, land use change, and fires should be responsible for 'anomalies' the dataset. Given that the Jena inversion depends strongly on modeled NPP products it's not surprising that is shows stronger 'trends' (see suggestion to de-trend data, above). Also, it would be interesting to see if fire fluxes were not backed out of the Jena inversion (again mentioned above) how the magnitude and timing of anomalies from these two data products compared to anomalies in the atmospheric $CO_2$ growth rate. This also could provide a better opportunities for the authors to illustrate the differences between the data products that are currently in the discussion.

Line 296 Carbon should be lowercase

The conclusion is really just a summary of results already presented (and repeated from the abstract). I'd omit this text, or say something more broadly about what we can infer from the study.

Fig. 6 & 8 I know abbreviations for each plant functional type are given in the text, but not using them in the caption or x-axis label bar make this figure hard to understand.

Fig. 6 Aren't there enough observations to include error estimates (or box-wisker plots) for Fluxnet sites?

Fig. 7 Caption and text should use the same (consistent) terminology here. I'm not really clear what is being compared here? How does one calculate a spatial correlation coefficient on two single values (e.g., correlation of IAV∼ mean temperature)?

Fig 7 The use of red-blue color bar on the left plots to show +/- correlation is confusing when on the right panels red-blue shows zonal mean correlations with trends or anomalies?

Fig 8 If this part of the analysis stays in the revised manuscript, I'd suggest the caption should be more descriptive (what are red and green bars).

Fig 9 I really don't understand what this figure is showing. The text & figure caption are not clear. More, the inset showing Western Europe seems strange. If this figure remains in the paper at all, would it make more sense to 1) omit the inset or 2) put it into supplementary material?

Fig 10 I also cannot understand I'm unclear what the color bar signifies (DRˆ2)? Is this the difference between TER/GPP when NEE < 0 during uptake periods and GPP/TER when NEE > 0 for MTE? If so, what does this difference of ratios really less us? I also still unclear how this is translated onto the Jena data?

————References—————

Le Quéré et al. 2014 Global carbon budget 2013, Earth Syst. Sci. Data, 6: 235–263,

doi:10.5194/essd-6-235-2014.

---

## Author Comment (AC1) · 10 Apr 2017

**Referee #1**

Major comments

1.  All three datasets are of dubious accuracy in representing interannual variability. The annual totals computed from eddy covariance sum much larger fluxes of opposing signs with likely systematic biases, especially in nighttime. The empirical upscaling was found to have relatively weak performance in representing interannual variability in a synthetic data experiment (without even accounting for any measurement or representativeness error in the training set) reported by Jung et al. (2009), for which the absence of soil moisture as a predictor is given by them as one reason. The inversion estimate, as the authors point out, is dominated at sub-continental scales by the (reasonable) prior assumption that variability scales with modeled NPP, and it probably contains little actual information from the CO2 time series at those scales. Could it makes sense to run the inversion with a more 'flat' prior, or a prior based on the MPI-MTE IAV, to get different IAV estimates?
    We agree with the reviewer that all the datasets used in the present analyses present weaknesses and lack of accuracy in representing the inter-annual variability. On the other side, this is what is currently available in terms of global-scale data of $CO_2$ land fluxes derived from inversions of atmospheric measurements or from the upscaling of surface flux observations. Following the reviewer suggestion, the limits of each product will be better discussed in the Materials and Methods section and strengths and weaknesses will be taken into account when evaluating results. Concerning the prior used for the Jena Inversion, it has indeed a seasonal pattern, however this is constant from one year to another, hence there is no influence of the prior on the IAV. The prior can only influence the fine-scale spatial pattern of IAV, since in the optimization the fluxes scale in space with the average prior flux. On the contrary the temporal IAV derives fully from the atmospheric signal. Using an MPI-MTE based prior for the Jena Inversion product would contaminate the IAV estimation, mostly because MPI-MTE varies in time, hence MPI-MTE IAV would influence the IAV derived from the Jena Inversion with the result that the two products wouldn't be independent any more.

2.  Figure 4 shows the dependence of median(?) IAV on resolution for the two gridded products. I wonder if something like this could be done with the available Fluxnet stations as well, for example with the help of a variogram (mean covariance of de-seasonalized NEE time series as a function of inter-station distance). This could help in deciding whether the lower IAV in the gridded products compared to Fluxnet is only because of the difference in spatial scale or is more intrinsic.
    We thank the reviewer for the interesting suggestion. We will add a new set of results to Fig 4 which will explore the dependence of IAV on the spatial averaging of the Fluxnet dataset, following the scheme used for the gridded product. The new series represents the IAV calculated from the Fluxnet database as a function of the area of aggregation of the sites, starting from single sites and then proceeding with averaging time series for groups of sites

located within an increasing distance. This procedure applied to flux sites mimics a decreasing resolution as done for the gridded products.

3. I didn't see any analysis of to what extent the IAV between the three products is actually in phase (i.e. the correlation of the deseasonalized NEE time series between the datasets). It would probably be relevant to show this.
We will consider this point together with point #1 raised by reviewer #3 and will perform an analysis on global averages of the two global products and of the Global Carbon Project estimates.

4. Also, forest inventories and crop yield statistics provide more reliable direct measurements of (at least above-ground) NPP and its IAV in many countries, potentially with rather good spatial coverage. Would there be any way to compare these to the IAV in the data sets reported here?
Following the suggestion of the reviewer we considered other possible data streams for the analysis but ultimately concluded that neither forest inventory nor yield statistics are appropriate for the present analysis. In fact, forest inventories are typically performed every 10-15 years, therefore they report NPP as a time average and for this reason they cannot be used in an inter-annual variability analysis. Crop yields are not necessary correlated to primary productivity, as they may be affected by events that do not affect GPP like for example a storm or frost at the end of the growing season that can fully compromise the yield but do not substantially change GPP.

Minor points

1. The element "carbon" is not capitalized (title and line 287).
The typo will be corrected.
2. Line 25: no comma before "that"
Comma will be cancelled
3. Figure 1c: It would be good to show the station network on the map.
Following the reviewer suggestion the station network will be plotted in Figure 1c
4. "Anomalies" sounds strange as a description of the IAV residuals from linear trend shown in Figure 5 and discussed in the text. Perhaps there is a better term.
As suggested by the reviewer we will use the term residuals.
5. The Jung et al. (2009) citation should be to the final paper, not the discussion paper.
The citation will be replaced with that of the final paper.
6. Formatting in the bibliography needs to be fixed, e.g. for Morgenstern et al. (2004) and others.
Bibliography will be checked and fixed.

**Referee #2**

General comments

1. There are some weaknesses. Some areas of the text, and a critical point or two in the methods, are unclear. Neither the MPI-MTE nor the inversion products seem ideal for this kind of IAV analysis, although I recognize that this is all there is to work with; still, the authors should address this.
   As stated by the reviewers the dataset used in the analysis are those available nowadays for the land $CO_2$ fluxes, namely i) site observations based on eddy covariance, ii) statistically upscaled products derived from site level measurements as MPI-MTE, or iii) inversion modeling products. We are aware of the weaknesses of the products used in this analysis and we plan to better discuss them together with their pros both in product descriptions and in the result discussion. Refer also to Referee #1 comment 1.

2. In addition, the conclusion should be re-done or removed; on a related note, the strengths and weaknesses of these NEE data products might be better, and more succinctly, summarized based on the analyses performed.
   We will prepare a new version of the conclusions following the suggestions of reviewer #2 and #3.

Specific comments

1. Lines 118-120: not as clear as it should be. Interannual variability computed with a 12-month window? How is this possible, as that's only 1 year?
   Analysis of IAV was based on the entire time series. Annual values were calculated not only for the "solar" years which were available in the dataset, but additional "years" were generated using a 12-month moving window which was shifted one month a time (Luyssaert et al. 2007).
2. L. 171-172: move to figure caption, or methods
   The sentence will be moved to Materials and Methods section 2.2
3. L. 197: "area of"
   The typo will be corrected
4. L. 241-243: unclear
   We will better clarify the concept in the revised text on the basis of what follows.
   The impact of climate drivers on IAV is based on a spatial analysis and not a temporal one. Spatial analyses of IAV in the inversion product are critical because at fine scale the spatial variability of the fluxes is mainly controlled by priors. In fact, the optimization algorithm of the inversion spatially allocates the fluxes proportionally to the prior; hence grid cells with higher productivity will change more if compared to cells with lower prior value (i.e. IAV at fine scale is proportional to the prior). For this reason we did not perform the spatial analysis on the inversion. On the contrary, prior does not affect the temporal analysis of IAV performed on the inversion product throughout the paper.
5. L. 250-: separating paragraphs, or indenting their first lines, would make this easier to read

Following the reviewer suggestion paragraph first lines were indented.

6. L. 286-: these aren't conclusions, just a recapitulation of results; remove

   As stated above we will reformulate the conclusions in the new version of the manuscript.

7. Figure 2: Rain (in axis title) or Precipitation (in caption)?

   Axis title will be modified in order to be consistent with the figure caption

**Referee #3**

General comments

1.  I feel like the paper is missing the bigger take home message I was looking for, to the globally (or Fluxnet) integrated anomalies in NEE match up with 1) each other and 2) anomalies in the land C sink the global carbon project (Le Quéré et al. 2014; these data are available in a downloadable spreadsheet at http://www.globalcarbonproject.org/carbonbudget)
    Even though the focus of the paper is on the pattern of IAV, we agree with the reviewer on the usefulness of a global inter-comparison of anomalies between products and with the GCP.
    In the new version of the manuscript we will therefore provide such a comparison, bearing in mind that GCP land fluxes are estimated as residual term from the atmospheric $CO_2$ budget and are therefore not completely independent from the inversion product.

2.  Since the paper is ostensibly about inter-annual variability in the terrestrial C cycle (NEE) what aren't all data products detrended first (these are weak responses anyway caused by different assumptions made with each approach)? Then the authors would be better able to address the IAV (or anomalies) which seem to be the focus of the paper.
    IAV is generally defined as the temporal variability of the annual flux as generated by trend and residuals (Yuan et al. 2009), for this reason in the manuscript we analyzed both components and quantified the relative magnitude of the two (e.g. Fig 5 show that IAV is dominated by the anomalies). We will make this clearer in the new version of the manuscript.

3.  What climate or weather data are used in MTE or the Jena inversion. Presumably neither used CRU (temperature) and GPCC (precipitation), as the authors of this paper chose to do? Thus, are analyses of climate drivers on IAV of NEE actually really just comparisons of distinct climate reanalysis products? Also, why not use the CRU precipitation product for consistency with the temperature data being used?
    MPI-MTE is based on the same climate drivers adopted in this analysis, namely CRU for temperature and GPCC for precipitation (Jung et al. 2011), while Jena-Inversion is not using any climate data in the flux calculation (with the exception of the wind field), being purely based on the atmospheric concentration measurements and an inversion transport model. GPCC precipitation was used instead of CRU for consistency with MPI-MTE, besides nowadays it is considered a better product as far as precipitation is concerned.

4.  Much of the text in section 3 is heavy on the results with little discussion and interpretation of the key findings. Although some sections do communicate broader statements about the findings (e.g. lines 197-206), similar thoughtful development of ideas should be included throughout this section
    In the revised version we will improve the discussion of results.

5. Why aren't correlations of IAV with site – level or global-scale climate drivers shown for Fluxnet or Jena inversion products?

The analysis of the global climate drivers of IAV was performed with the MPI-MTE because it is the only gridded product suitable for this purpose. The analysis has not been performed on the Jena Inversion products for the reasons explained in Reviewer #2 Specific Comments #4. Besides, a site level analysis is beyond the scope of the paper since it has already been performed in other papers (Luyssaert et al. 2007; Yuan et al. 2009; Wu et al. 2012).

6. I'm unclear what value is communicated by the calculation of CUP and CRP and would suggest removing these analyses from the paper. The finding that temperate and boreal systems have a stronger seasonal cycle in their CO2 drawdown seems obvious from atmospheric CO2 growth curves. Instead, if the purpose of these analysis is to "identify the role of photosynthesis and respiration as sources of IAV_NEE" (line 67), then it seems much more straightforward to just look at the IAV (or anomalies) of GPP and TER from the Fluxnet and MTE products directly. Then they could be correlated with climate drivers too? For example, at high latitudes do GPP and TER show strong temperature sensitivities, with anomalies GPP outpacing TER in warm years? Conversely, are Tropical GPP anomalies largely temperature related too, whereas TER shows less inter annual variability & climate sensitivity?

Since MPI-MTE and Fluxnet come from the same data source, while the Jena Inversion is a completely independent product, we think that it can be of interest to see if patterns are consistent. Since atmospheric inversion does not allow to separate GPP and TER, we used CUP and CRP as their proxies and we tested the validity of this assumption. Results of this analysis are shown in Fig 9, where we can infer that CUP and CRP are dominated by GPP and TER respectively. Although not being a precise GPP and TER estimation, $NEE_{CUP}$ and $NEE_{CRP}$ are highly correlated with them. We believe that the analysis of CUP and CRP brings additional information when performed on the inversion product. In particular at high latitudes where GPP/TER partitioning performed as CUP/CRP is particularly clear. Please refer also to point 13 of the Specific comments. In addition, the separation of the ecosystem $CO_2$ fluxes in these two terms is becoming increasingly common since it allows the description of a plant phenology based on carbon fluxes instead of greenness indexes.

Specific comments

1. I'm not used to seeing citations in the abstract. Is that the format for this journal?
   Citation will be removed.
2. I'm used to seeing ecosystem respiration referred to as ER, but maybe the authors are used to using different conventions?
   Both symbols can be found in the existing literature.
3. Line 55 This single paragraph is a single sentence consisting of a very long list of NEE estimation approaches. Why not break this into a sentence about each approach and discuss strengths/ weaknesses of each?

We take this point and will discuss further the strengths and weaknesses of the different approaches in the Materials/Methods sections.

4. Line 67 organization of objectives i), ii), and iii) don't align with the organization of methods and sections 3.1, 3.2 and 3.3. Can the objective reflect the broader layout of the paper?

   Objective order will be reorganized in accordance with the other sections of the manuscript.

5. I'd suggest Line 73 are "LaThuile and 2015" two distinct references?

   These are two subsequent releases of the Fluxnet dataset namely La Thuille and the 2015 releaese which are available at: http://fluxnet.fluxdata.org/data/la-thuile-dataset/ http://fluxnet.fluxdata.org/data/fluxnet2015-dataset/ We will add the links to the manuscript.

6. It's not clear if or how data were re-gridded (e.g. [1] subtracting finer scale RETRO and GFED4 fire fluxes from the Jena inversion, or [2] for temperature and precipitation in Fig. 2)

   Fire and Meteo Data were regridded using the aggregate function of the R-package raster, a sentence explaining this will be added to the manuscript.

7. More broadly, is subtracting for fire fluxes even necessary? Do the 14 observations extrapolated to this global product even 'see' the effect of forest fires? Don't the atmospheric inversion products the global carbon project implicitly see the effects of these fires? If so, why should they be subtracted out here?

   Inversion based estimates of land $CO_2$ fluxes include the signal of forest fires while MPI-MTE and Fluxnet don't. To maintain consistency in the analysis and to allow a proper comparison between products we decided to exclude fire driven IAV from the Jena Inversion product. We will make this clearer in the revised text.

8. Line 80. There are enough abbreviations in the text already. Are these needed too? Their use in lines 210-219 makes the text very hard to follow.

   Since the acronyms were only used in Fig 6, following the reviewer suggestion, they will be removed from the text and explained in Figure 6 caption.

9. Line 109 Air should not be capitalized.

   The typo will be corrected.

10. 'Jena inversion' or 'Jena Inversion' should be used consistently throughout the text.

    The spelling of the product name will be homogenized throughout the manuscript.

11. Were any lagged correlations explored to see if climate variability affected NEE in the subsequent season / year?

    Lagged correlations were beyond the scope of the paper. In the present paper only spatial patterns of the IAV dependence on climate drivers were analyzed.

12. Standard deviation and IAV are used interchangeably throughout the manuscript, but I think they mean the same thing? If so, just one term should be used for consistency. If they are different, it should be clarified in the text.

    The reviewer is correct the two terms can be used interchangeably to identify inter-annual variability as stated in Section 2.2 L1-2.

13. Line 133 I have no idea what this means "the difference between the two determination coefficients was computed" or where this analysis is presented (Fig 9)? More broadly, I'm unclear how / why the authors tried to infer something about GPP and TER from the inversion product.

We agree with the reviewer that the sentence was not clear. We will therefore improve the description of the analysis based on what follows.

Results of this analysis are presented in Figure 10 and 11. NEE was linearly correlated with GPP and TER (for Fluxnet sites and MPI-MTE, for which GPP and TER are available) and with $NEE_{CUP}$ (where CUP stands for Carbon Uptake Period), and $NEE_{CRP}$ (where CRP stands for Carbon release period) for the Jena Inversion product (for which GPP and TER are not available, and CUP and CRP were used as their proxies) to detect which of the two processes drives the IAV of NEE. The difference between the $R^2$ of the two regressions calculated for each pixel was plotted on maps in Fig 10 and in the climate space in Fig 11. The goodness of the assumption of CUP and CRP as proxies of GPP and TER was tested with the analysis shown in Fig 9.

14. Line 171. Why was IAV normalized using GPP estimates and not NEE, the later giving a real coefficient of variation (CV; grid cell standard deviation NEE / mean grid cell NEE). This should be clarified both in the text and caption. Also, shouldn't grid cell CV be calculated first, and then averaged over each climate bin?

IAV was not normalized with NEE because the latter fluctuates around zero and can lead to unreliably high values of CV. Normalization using GPP (which is always positive) offers a more robust metric of relative IAV.

The ratio of the means is a more robust estimation since mean of ratios is more sensible to outliers if compared to the ratio of the means, besides the latter gives more weight to points that bear more information.

We will clarify these methodological details in the revised document.

15. Line 180 & Fig. 3 I am unclear what insight this figure provides to the manuscript and it's sparingly discussed in the text. It's used to justify the CV calculation in Fig. 3 (line 173), but as this is a standard statistical approach I'm not sure it's warranted? As such, should the display item just be removed?

Fig 3 shows that in two datastreams (Fluxnet sites and Jena inversion) IAV increases monotonically and almost linearly with the productivity of the site. On the contrary MPI-MTE shows a different pattern, with a clear maximum followed by a decline of IAV in high productive sites. We think that this is due to the prominent role that FaPAR has in the MTE approach. Canopy greenness is particularly stable in the tropical humid forests (that are the most productive one) generating this unusual pattern of low relative IAV. We will discuss this aspect in further details in the revised version, since it is relevant to understand the general performance of the MTE model in the representation of the global IAV patterns.

16. Line 200. It seems like 'trends' in IAV should be driven mainly by environmental presses like atmospheric CO2 concentrations or broad-scale / chronic N deposition inputs. By contrast, climate variability, land use change, and fires should be responsible for 'anomalies' the dataset. Given that the Jena inversion depends strongly on modeled NPP products it's not surprising that is shows stronger 'trends' (see suggestion to detrend data, above). Also, it would be interesting to see if fire fluxes were not backed out of the Jena inversion (again mentioned above) how the magnitude and timing of anomalies from these two data products compared to anomalies in the atmospheric CO2 growth rate. This also could provide a better opportunities for the authors to illustrate the differences between the data products that are currently in the discussion.

As previously stated (Ref #1, General comment #1; Ref #2, Specific comment #4), the temporal dynamic of the land fluxes in the Jena inversion is totally driven by the atmospheric signal and fully independent from the prior, since the latter in this inversion scheme are time invariant. We argue that in the MTE product the lower IAV due to "trend" is due to the poor or lacking representation of environmental drivers like $CO_2$ or N deposition in this data product. We will make this clearer in the revised version.

17. Line 296 Carbon should be lowercase

   It will be corrected.

18. The conclusion is really just a summary of results already presented (and repeated from the abstract). I'd omit this text, or say something more broadly about what we can infer from the study.

   Conclusions will be reformulated in the new version of the manuscript as requested by the reviewers.

19. Fig. 6 & 8 I know abbreviations for each plant functional type are given in the text, but not using them in the caption or x-axis label bar make this figure hard to understand.

   Following the reviewer suggestion we will remove the acronyms from the text and we will adde their explanation to the figure caption.

20. Fig. 6 Aren't there enough observations to include error estimates (or box-wisker plots) for Fluxnet sites?

   Standard errors will be plotted for Fluxnet PFT IAV values.

21. Fig. 7 Caption and text should use the same (consistent) terminology here. I'm not really clear what is being compared here? How does one calculate a spatial correlation coefficient on two single values (e.g., correlation of IAV~ mean temperature)?

   The correlation was calculated in a moving spatial window of more than 600 points, we retrieved a IAV value and a temperature/precipitation value for each pixel. This will be better clarified in the revised text.

22. Fig 7 The use of red-blue color bar on the left plots to show +/- correlation is confusing when on the right panels red-blue shows zonal mean correlations with trends or anomalies?

   We take this point and will change the colors in the barplot to avoid misinterpretation of the figure.

23. Fig 8 If this part of the analysis stays in the revised manuscript, I'd suggest the caption should be more descriptive (what are red and green bars).

   Additional information will be added to the figure caption to make the figure more readable.

24. Fig 9 I really don't understand what this figure is showing. The text & figure caption are not clear. More, the inset showing Western Europe seems strange. If this figure remains in the paper at all, would it make more sense to 1) omit the inset or 2) put it into supplementary material?

   This figure will be better explained in the new version of the manuscript based on what follows. The aim of the figure is to highlight the role of GPP and TER (for MPI-MTE and Fluxnet) and of their proxy $NEE_{CUP}$ and $NEE_{CRP}$ (for Jena Inversion) in building up $IAV_{NEE}$. The determination coefficients were calculated for each pixel (for the gridded products) or site (for Fluxnet) fitting linear regressions of $IAV_{NEE}$ vs either GPP (TER), or $NEE_{CUP}$ ($NEE_{CRP}$). The difference in determination coefficients of GPP and TER linear regressions (the same holds for $NEE_{CUP}$ and

NEE$_{CRP}$) was used as a measure of which driver affects more IAV$_{NEE}$. Blue zones are GPP/CUP driven zones being the difference R$_{GPP}^2$-R$_{TER}^2$ (or R$_{CUP}^2$-R$_{CRP}^2$) positive while red zones are TER/CRP driven. See also Reviewer#3 answer #13. The inset was included in the graph because of the high site density of flux sites that characterizes Europe. Plotting an enlarged map allows in our opinion a better visualization of results.

25. Fig 10 I also cannot understand I'm unclear what the color bar signifies (DR^2)? Is this the difference between TER/GPP when NEE < 0 during uptake periods and GPP/TER when NEE > 0 for MTE? If so, what does this difference of ratios really less us? I also still unclear how this is translated onto the Jena data?
Figure 10 summarizes results plotted on maps in Figure 9 in a temperature/precipitation space. Blue pixels are GPP/CUP driven climate classes, red pixels are TER/CRP driven climate classes.

BIBLIOGRAPHY

Jung M, Reichstein M, Margolis H a., Cescatti A, Richardson AD, Arain MA, Arneth A, Bernhofer C, Bonal D, Chen J, Gianelle D, Gobron N, Kiely G, Kutsch W, Lasslop G, Law BE, Lindroth A, Merbold L, Montagnani L, Moors EJ, Papale D, Sottocornola M, Vaccari F, Williams C (2011) Global patterns of land-atmosphere fluxes of carbon dioxide, latent heat, and sensible heat derived from eddy covariance, satellite, and meteorological observations. J Geophys Res 116:G00J07. doi: 10.1029/2010JG001566

Luyssaert S, Janssens I a., Sulkava M, Papale D, Dolman  a. J, Reichstein M, Hollmén J, Martin JG, Suni T, Vesala T, Loustau D, Law BE, Moors EJ (2007) Photosynthesis drives anomalies in net carbon-exchange of pine forests at different latitudes. Glob Chang Biol 13:2110–2127. doi: 10.1111/j.1365-2486.2007.01432.x

Wu J, van der Linden L, Lasslop G, Carvalhais N, Pilegaard K, Beier C, Ibrom A (2012) Effects of climate variability and functional changes on the interannual variation of the carbon balance in a temperate deciduous forest. Biogeosciences 9:13–28. doi: 10.5194/bg-9-13-2012

Yuan W, Luo Y, Richardson AD, Oren R, Luyssaert S, Janssens I a., Ceulemans R, Zhou X, Grunwald T, Aubinet M, Berhofer C, Baldocchi DD, Chen J, Dunn AL, Deforest JL, Dragoni D, Goldstein AH, Moors E, William Munger J, Monson RK, Suyker AE, Starr G, Scott RL, Tenhunen J, Verma SB, Vesala T, Wofsy SC (2009) Latitudinal patterns of magnitude and interannual variability in net ecosystem exchange regulated by biological and environmental variables. Glob Chang Biol 15:2905–2920. doi: 10.1111/j.1365-2486.2009.01870.x

---

## Author Response (AR1)

**Referee #1**

Major comments

1. All three datasets are of dubious accuracy in representing interannual variability. The annual totals computed from eddy covariance sum much larger fluxes of opposing signs with likely systematic biases, especially in nighttime. The empirical upscaling was found to have relatively weak performance in representing interannual variability in a synthetic data experiment (without even accounting for any measurement or representativeness error in the training set) reported by Jung et al. (2009), for which the absence of soil moisture as a predictor is given by them as one reason. The inversion estimate, as the authors point out, is dominated at sub-continental scales by the (reasonable) prior assumption that variability scales with modeled NPP, and it probably contains little actual information from the CO2 time series at those scales. Could it makes sense to run the inversion with a more 'flat' prior, or a prior based on the MPI-MTE IAV, to get different IAV estimates?

   We agree with the reviewer that all the datasets used in the present analyses present weaknesses and lack of accuracy in representing the inter-annual variability. On the other hand, this is what is currently available in terms of global-scale data of $CO_2$ land fluxes derived from inversions of atmospheric measurements or from the upscaling of surface flux observations. Following the reviewer suggestion, the limits of each product were better discussed in the Materials and Methods section and strengths and weaknesses were taken into account when evaluating results.

   Concerning the prior used for the Jena Inversion, it has indeed a seasonal pattern, however this is constant from one year to another, hence there is no influence of the prior on the IAV. The prior can only influence the fine-scale spatial pattern of IAV, since in the optimization the fluxes scale in space with the average prior flux. On the contrary the temporal IAV derives fully from the atmospheric signal. Therefore in term of IAV running the inversion with a "flat" prior will not make any difference compared to the current analysis.

   On the contrary, using an MPI-MTE based prior for the Jena Inversion product would contaminate the IAV estimation, mostly because MPI-MTE varies in time, hence MPI-MTE IAV would influence the IAV derived from the Jena Inversion with the result that the two products wouldn't be independent any more.

2. Figure 4 shows the dependence of median(?) IAV on resolution for the two gridded products. I wonder if something like this could be done with the available Fluxnet stations as well, for example with the help of a variogram (mean covariance of de-seasonalized NEE time series as a function of inter-station distance). This could help in deciding whether the lower IAV in the gridded products compared to Fluxnet is only because of the difference in spatial scale or is more intrinsic.

   We thank the reviewer for the interesting suggestion. Following this advice we added new data-series to Fig 4 to explore the dependence of IAV on the spatial averaging of the Fluxnet dataset, following the scheme used for the gridded product. The new series represents the IAV

calculated from the Fluxnet database as a function of the area of aggregation of the sites, starting from single sites and then proceeding with averaging time series for groups of sites located within an increasing distance. This procedure applied to flux sites mimics a decreasing resolution as done for the gridded products.

3. I didn't see any analysis of to what extent the IAV between the three products is actually in phase (i.e. the correlation of the deseasonalized NEE time series between the datasets). It would probably be relevant to show this.
We have taken this point in consideration together with point #1 raised by reviewer #3 and performed an analysis on the temporal correlation between the global averages of the two global products and of the Global Carbon Project estimates. See new Figure 11 and related text.

4. Also, forest inventories and crop yield statistics provide more reliable direct measurements of (at least above-ground) NPP and its IAV in many countries, potentially with rather good spatial coverage. Would there be any way to compare these to the IAV in the data sets reported here?
Following the suggestion of the reviewer we considered other possible data streams for the analysis but ultimately concluded that neither forest inventory nor yield statistics are appropriate for the present analysis. In fact, forest inventories are typically performed every 10-15 years, therefore they report NPP as a time average and for this reason they cannot be used in an inter-annual variability analysis. On the other hand crop yields are not necessary correlated to primary productivity, as they may be affected by events that do not affect GPP like for example a storm or frost at the end of the growing season that can fully compromise the yield but do not substantially change GPP.

Minor points

1. The element "carbon" is not capitalized (title and line 287).
The typo was corrected.
2. Line 25: no comma before "that"
Comma was cancelled
3. Figure 1c: It would be good to show the station network on the map.
Following the reviewer suggestion the station network was plotted in Figure 1c
4. "Anomalies" sounds strange as a description of the IAV residuals from linear trend shown in Figure 5 and discussed in the text. Perhaps there is a better term.
As suggested by the reviewer we used the term residuals.
5. The Jung et al. (2009) citation should be to the final paper, not the discussion paper.
The citation was replaced with that of the final paper.
6. Formatting in the bibliography needs to be fixed, e.g. for Morgenstern et al. (2004) and others.
Bibliography was checked and fixed.

**Referee #2**

General comments

1. There are some weaknesses. Some areas of the text, and a critical point or two in the methods, are unclear. Neither the MPI-MTE nor the inversion products seem ideal for this kind of IAV analysis, although I recognize that this is all there is to work with; still, the authors should address this.

   As stated by the reviewers the dataset used in the analysis are those available nowadays for the land $CO_2$ fluxes, namely i) site observations based on eddy covariance, ii) statistically upscaled products derived from site level measurements as MPI-MTE, or iii) inversion modeling products. We are aware of the weaknesses of the products used in this analysis and we better discussed them together with their pros both in product descriptions and in the result discussion. Refer also to Referee #1 comment 1.

2. In addition, the conclusion should be re-done or removed; on a related note, the strengths and weaknesses of these NEE data products might be better, and more succinctly, summarized based on the analyses performed.

   Following the suggestions of reviewer #2 and #3 we wrote two new paragraphs at the end of the Result and Discussion section as Conclusions.

Specific comments

1. Lines 118-120: not as clear as it should be. Interannual variability computed with a 12-month window? How is this possible, as that's only 1 year?

   Analysis of IAV was based on the entire time series. Annual values were calculated not only for the "solar" years which were available in the dataset, but additional "years" were generated using a 12-month moving window which was shifted one month a time, following the methodology proposed by Luyssaert et al. (2007).

2. L. 171-172: move to figure caption, or methods

   The sentence was moved to Materials and Methods section 2.2

3. L. 197: "area of"

   The typo was corrected

4. L. 241-243: unclear

   We better clarified the concept in the revised text on the basis of what follows.

   The impact of climate drivers on IAV is based on a spatial analysis and not a temporal one. Spatial analyses of IAV in the inversion product are critical because at fine scale the spatial variability of the fluxes is mainly controlled by priors. In fact, the optimization algorithm of the inversion spatially allocates the fluxes proportionally to the prior; hence grid cells with higher productivity will change more if compared to cells with lower prior value (i.e. IAV at fine scale is proportional to the prior). For this reason we did not perform the spatial analysis on the inversion. On the contrary, prior does not affect the temporal analysis of IAV performed on the inversion product throughout the paper.

5. L. 250-: separating paragraphs, or indenting their first lines, would make this easier to read
   Following the reviewer suggestion paragraph first lines were indented.
6. L. 286-: these aren't conclusions, just a recapitulation of results; remove
   As stated above we reformulated the conclusions in the new version of the manuscript.
7. Figure 2: Rain (in axis title) or Precipitation (in caption)?
   Axis title was modified in order to be consistent with the figure caption

**Referee #3**

General comments

1. I feel like the paper is missing the bigger take home message I was looking for, to the globally (or Fluxnet) integrated anomalies in NEE match up with 1) each other and 2) anomalies in the land C sink the global carbon project (Le Quéré et al. 2014; these data are available in a downloadable spreadsheet at http://www.globalcarbonproject.org/carbonbudget)
Even though the focus of the paper is on the pattern of IAV, we agree with the reviewer on the usefulness of a global inter-comparison of anomalies between products and with the GCP.
In the new version of the manuscript we therefore provided such a comparison, bearing in mind that GCP land fluxes are estimated as residual term from the atmospheric $CO_2$ budget and are therefore not completely independent from the inversion product.

2. Since the paper is ostensibly about inter-annual variability in the terrestrial C cycle (NEE) what aren't all data products detrended first (these are weak responses anyway caused by different assumptions made with each approach)? Then the authors would be better able to address the IAV (or anomalies) which seem to be the focus of the paper.
IAV is generally defined as the temporal variability of the annual flux as generated by trend and residuals (Yuan et al. 2009), for this reason in the manuscript we analyzed both components and quantified the relative magnitude of the two (e.g. Fig 5 show that IAV is dominated by the anomalies). We made this clearer in the new version of the manuscript.

3. What climate or weather data are used in MTE or the Jena inversion. Presumably neither used CRU (temperature) and GPCC (precipitation), as the authors of this paper chose to do? Thus, are analyses of climate drivers on IAV of NEE actually really just comparisons of distinct climate reanalysis products? Also, why not use the CRU precipitation product for consistency with the temperature data being used?
MPI-MTE is based on the same climate drivers adopted in this analysis, namely CRU for temperature and GPCC for precipitation (Jung et al. 2011), while Jena-Inversion is not using any climate data in the flux calculation (with the exception of the wind field), being purely based on the atmospheric concentration measurements and an inversion transport model. GPCC precipitation was used instead of CRU for consistency with MPI-MTE, besides nowadays it is considered a better product as far as precipitation is concerned.

4. Much of the text in section 3 is heavy on the results with little discussion and interpretation of the key findings. Although some sections do communicate broader statements about the findings (e.g. lines 197-206), similar thoughtful development of ideas should be included throughout this section
Following the advice of referee 2 and 3 in the revised version we expanded the discussion of the results presented in each figure.

5. Why aren't correlations of IAV with site – level or global-scale climate drivers shown for Fluxnet or Jena inversion products?

   The analysis of the global climate drivers of IAV was performed with the MPI-MTE because it is the only gridded product suitable for this purpose. The analysis has not been performed on the Jena Inversion products for the reasons explained in Reviewer #2 Specific Comments #4. Besides, a site level analysis is beyond the scope of the paper since it has already been addressed in other papers (Luyssaert et al. 2007; Yuan et al. 2009; Wu et al. 2012).

6. I'm unclear what value is communicated by the calculation of CUP and CRP and would suggest removing these analyses from the paper. The finding that temperate and boreal systems have a stronger seasonal cycle in their CO2 drawdown seems obvious from atmospheric CO2 growth curves. Instead, if the purpose of these analysis is to "identify the role of photosynthesis and respiration as sources of IAV_NEE" (line 67), then it seems much more straightforward to just look at the IAV (or anomalies) of GPP and TER from the Fluxnet and MTE products directly. Then they could be correlated with climate drivers too? For example, at high latitudes do GPP and TER show strong temperature sensitivities, with anomalies GPP outpacing TER in warm years? Conversely, are Tropical GPP anomalies largely temperature related too, whereas TER shows less inter annual variability & climate sensitivity?

   Since MPI-MTE and Fluxnet come from the same data source, while the Jena Inversion is a completely independent product, we think that it can be of interest to see if IAV patterns are consistent between products. Since atmospheric inversion does not allow to separate GPP and TER, we used CUP and CRP as their proxies and we tested the validity of this assumption. Results of this analysis are shown in Fig 9, where we can infer that CUP and CRP are dominated by GPP and TER respectively. Although not being perfect GPP and TER proxies, $NEE_{CUP}$ and $NEE_{CRP}$ are highly correlated with them. We believe that the analysis of CUP and CRP brings additional information when performed on the inversion product. In particular at high latitudes where GPP/TER partitioning performed as CUP/CRP is particularly clear. Please refer also to point 13 of the Specific comments. In addition, the separation of the ecosystem $CO_2$ fluxes in these two terms is becoming increasingly common since it allows the description of a plant phenology based on carbon fluxes instead of greenness indexes.

Specific comments

1. I'm not used to seeing citations in the abstract. Is that the format for this journal?
   Citation was removed.
2. I'm used to seeing ecosystem respiration referred to as ER, but maybe the authors are used to using different conventions?
   Both acronyms are used in the literature on the topic.
3. Line 55 This single paragraph is a single sentence consisting of a very long list of NEE estimation approaches. Why not break this into a sentence about each approach and discuss strengths/ weaknesses of each?

We take this point and discussed further the strengths and weaknesses of the different approaches in the Materials/Methods sections.

4. Line 67 organization of objectives i), ii), and iii) don't align with the organization of methods and sections 3.1, 3.2 and 3.3. Can the objective reflect the broader layout of the paper?

   Objective order was reorganized in accordance with the other sections of the manuscript.

5. I'd suggest Line 73 are "LaThuile and 2015" two distinct references?

   These are two subsequent releases of the Fluxnet dataset namely La Thuille and the 2015 releaese which are available at: http://fluxnet.fluxdata.org/data/la-thuile-dataset/ http://fluxnet.fluxdata.org/data/fluxnet2015-dataset/ We added the links to the manuscript.

6. It's not clear if or how data were re-gridded (e.g. [1] subtracting finer scale RETRO and GFED4 fire fluxes from the Jena inversion, or [2] for temperature and precipitation in Fig. 2)

   Both fire and meteo data were regridded using the aggregate function of the R-package raster, a sentence explaining this was added to the manuscript.

7. More broadly, is subtracting for fire fluxes even necessary? Do the 14 observations extrapolated to this global product even 'see' the effect of forest fires? Don't the atmospheric inversion products the global carbon project implicitly see the effects of these fires? If so, why should they be subtracted out here?

   Inversion based estimates of land $CO_2$ fluxes include the signal of forest fires while MPI-MTE and Fluxnet don't. To maintain consistency in the analysis and to allow a proper comparison between products we decided to exclude fire driven IAV from the Jena Inversion product. We made this clearer in the revised text.

8. Line 80. There are enough abbreviations in the text already. Are these needed too? Their use in lines 210-219 makes the text very hard to follow.

   Since the acronyms were only used in Fig 6, following the reviewer suggestion, they were removed from the text and explained in Figure 6 caption.

9. Line 109 Air should not be capitalized.

   The typo was corrected.

10. 'Jena inversion' or 'Jena Inversion' should be used consistently throughout the text.

    The spelling of the product name was homogenized throughout the manuscript.

11. Were any lagged correlations explored to see if climate variability affected NEE in the subsequent season / year?

    Despite potentially interesting, lagged correlations between drivers and IAV were beyond the scope of the paper. In the present paper only spatial patterns of the IAV dependence on climate drivers were analyzed.

12. Standard deviation and IAV are used interchangeably throughout the manuscript, but I think they mean the same thing? If so, just one term should be used for consistency. If they are different, it should be clarified in the text.

    The reviewer is correct the two terms can be used interchangeably to identify inter-annual variability as stated in Section 2.2 L1-2.

13. Line 133 I have no idea what this means "the difference between the two determination coefficients was computed" or where this analysis is presented (Fig 9)? More broadly, I'm

unclear how / why the authors tried to infer something about GPP and TER from the inversion product.

We agree with the reviewer that the sentence was not clear. We therefore improved the description of the analysis based on what follows.

Results of this analysis are presented in Figure 10 and 11. NEE was linearly correlated with GPP and TER (for Fluxnet sites and MPI-MTE, for which GPP and TER are available) and with $NEE_{CUP}$ (where CUP stands for Carbon Uptake Period), and $NEE_{CRP}$ (where CRP stands for Carbon release period) for the Jena Inversion product (for which GPP and TER are not available, and CUP and CRP were used as their proxies) to detect which of the two processes drives the IAV of NEE. The difference between the R2 of the two regressions calculated for each pixel was plotted on maps in Fig 10 and in the climate space in Fig 11. The goodness of the assumption of CUP and CRP as proxies of GPP and TER was tested with the analysis shown in Fig 9.

14. Line 171. Why was IAV normalized using GPP estimates and not NEE, the later giving a real coefficient of variation (CV; grid cell standard deviation NEE / mean grid cell NEE). This should be clarified both in the text and caption. Also, shouldn't grid cell CV be calculated first, and then averaged over each climate bin?

IAV was not normalized with NEE because the latter fluctuates around zero and can lead to unreliably high values of CV and to both positive and negative values. Normalization using GPP (which is always positive and significantly larger than zero) offers a more robust metric of relative IAV.

We use the ratio of the means because it is a more robust metric since mean of ratios is more sensitive to outliers if compared to the ratio of the means, besides the latter gives more weight to points that bear more information.

We clarified these methodological details in the revised document.

15. Line 180 & Fig. 3 I am unclear what insight this figure provides to the manuscript and it's sparingly discussed in the text. It's used to justify the CV calculation in Fig. 3 (line 173), but as this is a standard statistical approach I'm not sure it's warranted? As such, should the display item just be removed?

Fig 3 shows that in two data-streams (Fluxnet sites and Jena inversion) IAV increases monotonically and almost linearly with the productivity of the site. On the contrary MPI-MTE shows a different pattern, with a clear maximum followed by a decline of IAV in high productive sites. We think that this is due to the prominent role that FaPAR has in the MTE approach. Canopy greenness is particularly stable in the tropical humid forests (that are the most productive one) generating this unusual pattern of low relative IAV. We discussed this aspect in further details in the revised version, since it is relevant to understand the general performance of the MTE model in the representation of the global IAV patterns.

16. Line 200. It seems like 'trends' in IAV should be driven mainly by environmental presses like atmospheric CO2 concentrations or broad-scale / chronic N deposition inputs. By contrast, climate variability, land use change, and fires should be responsible for 'anomalies' the dataset. Given that the Jena inversion depends strongly on modeled NPP products it's not surprising that is shows stronger 'trends' (see suggestion to detrend data, above). Also, it would be interesting to see if fire fluxes were not backed out of the Jena inversion (again mentioned above) how the

magnitude and timing of anomalies from these two data products compared to anomalies in the atmospheric CO2 growth rate. This also could provide a better opportunities for the authors to illustrate the differences between the data products that are currently in the discussion.

As previously stated (Ref #1, General comment #1; Ref #2, Specific comment #4), the temporal dynamic of the land fluxes in the Jena inversion is totally driven by the atmospheric signal and fully independent from the prior, since the latter in this inversion scheme is time invariant. We argue that in the MTE product the lower IAV due to "trend" is due to the poor or lacking representation of environmental drivers like $CO_2$ or N deposition in this data product. We made this clearer in the revised version.

17. Line 296 Carbon should be lowercase

It was corrected.

18. The conclusion is really just a summary of results already presented (and repeated from the abstract). I'd omit this text, or say something more broadly about what we can infer from the study.

Conclusions were reformulated in the new version of the manuscript as requested by the reviewers.

19. Fig. 6 & 8 I know abbreviations for each plant functional type are given in the text, but not using them in the caption or x-axis label bar make this figure hard to understand.

Following the reviewer suggestion we removed the acronyms from the text and we added their explanation to the figure caption.

20. Fig. 6 Aren't there enough observations to include error estimates (or box-wisker plots) for Fluxnet sites?

Following the suggestion of the reviewer standard errors were plotted for Fluxnet PFT IAV values.

21. Fig. 7 Caption and text should use the same (consistent) terminology here. I'm not really clear what is being compared here? How does one calculate a spatial correlation coefficient on two single values (e.g., correlation of IAV~ mean temperature)?

The correlation was calculated in a moving spatial window of more than 600 points, we retrieved a IAV value and a temperature/precipitation value for each pixel. This was better clarified in the revised text.

22. Fig 7 The use of red-blue color bar on the left plots to show +/- correlation is confusing when on the right panels red-blue shows zonal mean correlations with trends or anomalies?

We take this point and changed the colors in the barplot to avoid misinterpretation of the figure.

23. Fig 8 If this part of the analysis stays in the revised manuscript, I'd suggest the caption should be more descriptive (what are red and green bars).

Additional information was added to the figure caption to make the figure more readable.

24. Fig 9 I really don't understand what this figure is showing. The text & figure caption are not clear. More, the inset showing Western Europe seems strange. If this figure remains in the paper at all, would it make more sense to 1) omit the inset or 2) put it into supplementary material?

This figure was better explained in the new version of the manuscript based on what follows.

The aim of the figure is to highlight the role of GPP and TER (for MPI-MTE and Fluxnet) and of their proxy $NEE_{CUP}$ and $NEE_{CRP}$ (for Jena Inversion) in building up $IAV_{NEE}$. The determination coefficients were calculated for each pixel (for the gridded products) or site (for Fluxnet) fitting linear regressions of $IAV_{NEE}$ vs either GPP (TER), or $NEE_{CUP}$ ($NEE_{CRP}$). The difference in determination coefficients of GPP and TER linear regressions (the same holds for $NEE_{CUP}$ and $NEE_{CRP}$) was used as a measure of which driver affects more $IAV_{NEE}$. Blue zones are GPP/CUP driven zones being the difference $R_{GPP}^2 - R_{TER}^2$ (or $R_{CUP}^2 - R_{CRP}^2$) positive while red zones are TER/CRP driven. See also Reviewer#3 answer #13. The inset was included in the graph because of the high site density of flux sites that characterizes Europe. Plotting an enlarged map allows in our opinion a better visualization of results.

25. Fig 10 I also cannot understand I'm unclear what the color bar signifies (DRˆ2)? Is this the difference between TER/GPP when NEE < 0 during uptake periods and GPP/TER when NEE > 0 for MTE? If so, what does this difference of ratios really less us? I also still unclear how this is translated onto the Jena data?

    Figure 10 summarizes results plotted on maps in Figure 9 in a temperature/precipitation space. Blue pixels are GPP/CUP driven climate classes, red pixels are TER/CRP driven climate classes.

[revised manuscript text omitted]

Figure 11 reports the results of the comparison of the two global products used in the present analysis with the Global Carbon Project dataset. The right panel shows the time series of annual anomalies of the three products. Jena Inversion shows a good agreement with the GCP, and this is not surprising since GCP land fluxes are estimated as residual term from the atmospheric $CO_2$ budget and are therefore are not completely independent from the Jena Inversion product. On the other hand, this analysis highlights the well-known limits of the MPI-MTE product in representing the IAV.

**4 Conclusions**

Patterns and controls of the inter annual variability of Ccarbon net ecosystem exchange have been investigated using three different datasets: ecosystem-level data from the FLUXNET database, the MPI-MTE bottom-up statistical upscaling of surface fluxes, and a top-down product based on atmospheric concentration data (Jena CarboScope $CO_2$ inversion).

The global average of site-level $IAV_{NEE}$ ($\sim$130 $gC$ $m^{-2}y^{-1}$), computed as the standard deviation of annual NEE, was observed to be almost 6 times the values calculated from the two global products (15 and 20 $gC$ $m^{-2}$ $y^{-1}$ for MPI-MTE and Jena Inversion, respectively). This difference is probably due to the large variability in the spatial scale of point-level and gridded products, combined with the scale dependence of the IAV signal, as shown in Fig 4 for the gridded products.

All datasets exhibited smaller IAV at higher latitudes, whereas arid ecosystems showed the largest IAV in the global products. Temperature has the highest correlation with the spatial patterns of IAV, with a positive control at temperature-limited northern ecosystems and a negative control in water limited zones. Further insights in the sources of IAV have been achieved by exploring the temporal variability of the two gross components: GPP and TER. NEE fluxes during the carbon uptake and carbon release period were used as proxies of GPP and TER, respectively, since the partitioned fluxes were not available for the Jena Inversion. In all three datasets, GPP and $NEE_{CUP}$, respectively, were shown to control consistently the inter annual variability NEE across geographical and climate domains, highlighting the fundamental role of photosynthesis in driving the temporal fluctuation of the land sink.

[revised manuscript text omitted]

---

## Referee Report (RR1)

The authors responded well to the reviewer comments, and I think the paper is essentially suitable for publication. I noticed the following minor corrections on re-reading the paper:

l. 103: I would not call this "measurement errors"
104-105: "which cannot be assessed" – an overstatement, considering that there is an entire TransCom project that has been devoted to doing exactly that
128: good to give a citation for the R package
334: ; –> ,
340: "barely correlated" – please also give a numerical value
346: were –> where

---

## Author Response (AR2)

**Reviewer 1**

The authors responded well to the reviewer comments, and I think the paper is essentially suitable for publication. I noticed the following minor corrections on re-reading the paper:

103: I would not call this "measurement errors"
Following the suggestion we replaced the definition with "incompleteness of the accounted fluxes"
104-105: "which cannot be assessed" — an overstatement, considering that there is an entire TransCom project that has been devoted to doing exactly that
We agree with the comment of the reviewer and have changed the sentence accordingly in addition to adding the following reference to TransCom.
Baker, D. F. et al. TransCom 3 inversion intercomparison: Impact of transport model errors on the interannual variability of regional CO2 fluxes, 1988-2003. Global Biogeochem. Cycles 20, 1988–2003 (2006).
128: good to give a citation for the R package
A citation was added.
334: ; —> ,
The change was made.
340: "barely correlated" — please also give a numerical value
R2 and p values were added.
346: were —> where
The typo was corrected.

**Reviewer 2**

Marcolla and co-authors have done a nice job revising the manuscript. In particular, I appreciate the efforts that went into interpreting the results in section 3 of the revised text.

Broad comments
Sorry to keep harping on this, but if IAV is the SD of NEE calculated using a sliding, 12-month window (line 130) doesn't this effectively remove potential trends in the results? This isn't a field I work in, but results in Fig 5 showing that the 'residuals' explain nearly all of the variance in NEE seems like a completely expected result of how the analysis was done. Perhaps this assumption on my part is misconstrued, but I'd recommend removing this figure from the text, as it does little to inform broader message of the manuscript. More, the regions where trends (line 238) seem more important in the Jena product seem particularly void of observational constraints (Fig 1c).

We confirm that the results presented in Fig. 5 are not an artifact produced by the moving window. To support this statement we have reproduced the same figure using the yearly values (jan-dec) of NEE (see figure below) and the results are almost identical to those produced with the moving window approach. This latter method is typically used in the literature to avoid any subjective choice in the definition of the "year", which is particularly relevant in the tropics and in the southern hemisphere, where a jan-dec windows clearly does not match the annual cycle of ecosystems.
Given that the moving-window method does not generate any artifact, we prefer to maintain the original version of Fig. 5. In addition, this figure is bringing interesting information about the possible underestimation of the trend driven IAV in MTE. In general the figure highlights the limited importance of the trend in the IAV, showing that in the last 30 years climate variability more than climate change has dominated the yearly fluctuation of the terrestrial carbon budget.

[Figure]

Figure 5 data analysis performed without the moving window

[Figure]

Figure 5 as reported in the manuscript with the moving window

Minor comments & Technical corrections

Again, standard deviation and IAV are used interchangeably throughout the manuscript, since they mean the same thing please just use one term and stick with it throughout the manuscript.
Following the suggestion of the reviewer the term "standard deviation" was replaced by IAV throughout the manuscript when appropriate.

Line 17. I might reorganize this sentence for clarity: "Results show that the global average of IAV, quantified as the standard deviation of annual NEE, at FLUXNET sites is ~120 g C m-2 y-1 and peaks in arid ecosystems. This variability is almost six times larger than…"
Following the reviewer suggestion, the sentence was reorganized.

Line 20. It's unclear what the "two data-driven global products" are referring to (I'm assuming this is MPI and Jena)? If so maybe this sentence can be reorganized, "Most of the temporal variability observed in the last three decades of the MPI-MTE and Jena Inversion products is due to yearly anomalies, whereas the temporal trends explain only about 15% and 20% of the variability, respectively.
The sentence was replaced with the one suggested by the reviewer.

Why is tropics often capitalized?
We changed into lowercase.

Line 92, can a reference be supplied for this statement, or is all in the 2011 Jung paper?
Yes, it all refers to Jung et al, 2011.

I can see investigating the lag correlations of climate drivers and NEE variability is perhaps outside the scope of the paper, but still feel the topic still should be mentioned, as ground observations clearly demonstrate a temporal mismatch between the timing of weather anomalies and their ultimate influence on different components of the C cycle (e.g. Doughty et al. 2015). The lack of potential lags is mentioned for the MTE product (line 95, but not the present study)

We agree with the referee that lag effects can influence the temporal dependence of the terrestrial carbon budget on the climate drivers. On the other hand the present study focuses mostly on the role of GPP/TER and/or CUP/CRP in determining the inter-annual variability in NEE, hence both the dependent and independent variable involved in the analyses are lagged compared to climate.

The issue has a larger importance when mentioning the missing representation of time lags in the MPI-MTE, since climate drivers are used to predict the fluxes at global scale and hence ignoring the presence of lags can indeed affect the flux estimations, as stressed by the referee.

We have added a sentence in the section 3.2 to address this point and we have added the reference suggested by the referee.

Line 136, maybe it's explained elsewhere, but how were IAV[NEE] normalized?

Since it was firstly explained in the results section we added a sentence on the normalization of IAV at the line indicated by the reviewer.

Line 161 "resumed" is a funny word choice here. Maybe replace with 'shown'

We replaced "resumed" with "shown".

Line 204. What is shown in the right panels? The text says "it's the ratio of the mean IAV and GPP (right column)" while the caption claims "IAV[NEE] (CV[NEE], right panels). Please be clear and consistent with language throughout the text.

Both in the text and in figure 2 caption we refer to normalized IAV. In the figure caption we introduced the symbol $CV_{NEE}$ to explain the axis label.

Line 239. Please provide references to support this claim.

The following reference was added:

Forkel, M. et al. Enhanced seasonal $CO_2$ exchange caused by amplified plant productivity in northern ecosystems. Science (80). 351, 696–699 (2016).

Lines 250 to 265. The eruption of undefined pft abbreviations in this paragraph make the text nearly unintelligible.

We explained the acronyms to make the sentence clearer.

Please refer to particular panels in the figure captions.

Fig 1 Would marking the observations used in the MTE product be useful (as in panels A & B)? Also, the figure caption should call out the different scale bars used for FLUXNET and gridded results.

Most of the observations used in the MTE products are actually plotted in panel (a), superimpose the points in panel b would completely cover the map colors especially in Europe. A sentence was added to the caption which highlights the different scales used.

Fig. 2. Please clarify in the caption what color bar is being used for MPI IAV. I'm assuming it's the same as the Jena panels below, but it's not really clear.

Following the reviewer suggestion it was specified in the caption.

Fig. 3. I don't see any black dots on the figure, I'm assuming MPI is in grey?
Corrected.

Fig. 4. It's confusing to have the dot colors switch between data products in Figs 3 & 4. Can these display items use a common color scheme (e.g. fluxnet = red, Jena = green). Selecting colors appropriate for color blind readers would also be nice.
Colors were made consistent in Fig 3 and 4, and turned into colors suitable for color blind readers.

Fig 8 referring to panels in the caption would be helpful. Also, avoiding abbreviations in the caption will help reader understanding.
References to the figure panels were added to the figure caption.

Figure 9 is still unclear. Is this the grid cell correlation between IAV[NEE] and (a) MPI GPP-TER (b) MIP CUP – CRP and (c) Jena CUP – CRP? Again, the caption is not clear.
Analytical expressions of the plotted variables were added to the figure caption to improve readability.

References:
Doughty, C. E., et al. (2015), Drought impact on forest carbon dynamics and fluxes in Amazonia, Nature, 519(7541), 78-82, doi:10.1038/nature14213.